# Gray matter correlates of childhood maltreatment lack replicability in a multi-cohort brain-wide association study

Janik Goltermann [1] ✉, Nils R. Winter [1], Susanne Meinert [1,2], Dominik Grotegerd[1], Anna Kraus [1], Kira Flinkenflügel [1], Luisa Altegoer[1], Judith Krieger[1], Elisabeth J. Leehr[1], Joscha Böhnlein [1,3], Linda M. Bonnekoh[1,4], Maike Richter[1,5], Tim Hahn [1], Lukas Fisch[1], Marius Gruber[1,6,7], Marco Hermesdorf[8], Klaus Berger[8], Volker Arolt [1], Katharina Brosch[9,10], Frederike Stein[9], Florian Thomas-Odenthal [9], Paula Usemann[9], Lea Teutenberg[9], Vincent Hammes[9], Hamidreza Jamalabadi [9], Nina Alexander[9], Benjamin Straube [9], Andreas Jansen[9], Igor Nenadić [9], Tilo Kircher [9], Nils Opel [5,11,13] & Udo Dannlowski [1,12,13]

Childhood maltreatment effects on cerebral gray matter have been frequently discussed as a neurobiological pathway for depression. However, localizations are highly heterogeneous, and recent reports have questioned the replicability of mental health neuroimaging findings. Here, we investigate the replicability of gray matter correlates of maltreatment, measured retrospectively via the Childhood Trauma Questionnaire, across three large adult cohorts (total N = 3225). Pooling cohorts yields maltreatment-related gray matter reductions, with most extensive effects when not controlling for depression diagnosis (maximum partial $R^2$ = .022). However, none of these effects significantly replicate across cohorts. Non-replicability is consistent across a variety of maltreatment subtypes and operationalizations, as well as subgroup analyses with and without depression, and stratified by sex. Results are furthermore consistent across a variety of gray matter operationalizations, including voxel-based morphometry and parcellation-based cortical and subcortical measures. In this work, we show that there is little evidence for the replicability of gray matter correlates of childhood maltreatment, when adequately controlling for psychopathology. This underscores the need to focus on replicability research in mental health neuroimaging.

Childhood maltreatment (CM) has been identified to be one of the most important risk factors for the development of affective disorders[1,2] and is associated with chronic disease trajectories and poorer treatment outcomes in major depressive disorder (MDD)[1,3]. Within the past two decades, a plethora of neuroimaging studies has repeatedly suggested that experiences of abuse and neglect during childhood are associated with neurobiological alterations in adults[4–9]. Brain regions where these effects have been localized overlap with neural correlates of MDD, giving rise to the notion that neurobiological alterations may mediate the unfavorable effects of CM on clinical trajectories[10,11]. Thus, studying the neurobiological correlates of CM could give insights into the mechanistic processes of its clinical

consequences, potentially informing the optimization of treatments or preventative measures for this population[12].

In adults, CM effects on gray matter structure have been observed in an array of regions, with most frequent findings implying the hippocampus, amygdala, dorsolateral prefrontal cortex, insula and anterior cingulate cortex[9,13–15]. However, the investigation of CM-associated gray matter alterations has yielded considerable heterogeneity in findings regarding the localization of effects. Importantly, large-scale consortium studies and meta-analyses do not find these aforementioned regions, but rather report a multitude of other areas to be associated with CM, including the postcentral gyrus and occipital regions[14], the median cingulate gyri and supplementary motor area[16], the cerebellum and striatum[17], as well as the precuneus[18].

This heterogeneity could result from the diversity of measurement instruments (e.g., different retrospective self-report scales vs. prospective ratings) and operationalizations of CM (e.g., continuous vs. categorical), as well as different subtypes of maltreatment being studied separately (e.g., Ringwald et al.[19], Sheffield et al.[20], Van Harmelen et al.[21]). Regarding subtypes of CM, there has been considerable debate whether neural correlates could be specific to individual types of experiences. The increasingly influential dimensional model of adversity postulates that different dimensions of CM, such as threat-related and deprivation-related experiences or the unpredictability of one's environment, underly differential neurobiological processes, consequently leading to differential neural correlates[22,23]. Evidence for this model in children and adolescents has been accumulated over several studies[6]. In contrast, other scholars have suggested the relevance of dividing CM experiences even further and have argued that brain alterations are aligned to these experiences in a very specific manner, such as parental verbal abuse impacting gray matter within the auditory cortex or sexual abuse being associated with cortical thinning within the somatosensory cortex[9]. Another potential source of heterogeneity could stem from varying sample characteristics, differing in diagnoses, the degree of psychopathology and the severity of CM exposure (e.g., McLaughlin et al.[24]). Furthermore, different statistical approaches have been used. One statistical challenge is a strong phenomenological co-occurrence with mental health problems. Often, psychiatric diagnosis is statistically controlled for, which leads to reduced power to detect maltreatment effects because both constructs strongly covary[1] and both explain shared variance in neurobiological alterations[10]. On the other side, if not controlling for diagnosis, neurobiological effects due to maltreatment or due to depression are impossible to disentangle. Moreover, evidence suggests that the neural correlates of CM may differ by sex[25–27] and could be moderated by age[18], underscoring the importance of carefully considering these factors in investigations of CM effects.

The recent debate around questionable replicability in the neuroimaging domain due to underpowered samples and publication bias suggests the possibility of substantial false-positive findings within the previous body of evidence[28,29]. This notion is supported by evidence for considerable publication bias in the meta-analyzed findings of gray matter correlates of CM[13]. In fact, large-scale neuroimaging consortia, such as the ENIGMA consortium (Frodl et al.[26], $n = 3036$) or the UK-Biobank (Gheorghe et al.[17], $n = 6751$), have yielded much smaller effect sizes compared to studies with smaller samples, and have failed to replicate frequently reported associations of CM with the hippocampus or amygdala. However, these consortia still rely exclusively on segmented volumetric brain measures, thus losing spatial resolution, which may account for lower sensitivity to find gray matter alterations, posing a limitation to these findings.

In summary, inconclusive previous findings may result from variability in CM operationalizations, investigated clinical and non-clinical subgroups, varying statistical approaches, insufficient spatial resolution or simply because of false-positive results originating from underpowered studies. Systematic investigations of the replicability of

these neural correlates do not exist to date. To shed light on this heterogeneity and re-evaluate our knowledge about the neurobiological underpinnings of adverse childhood experiences, we investigated the cross-cohort replicability of gray matter correlates of CM. We therefore utilized three large-scale, deeply phenotyped clinical cohort datasets, with a broad range of self-reported maltreatment experiences, in combination with high-resolution voxel-based morphometry (VBM). These rich datasets were assessed and processed in standardized pipelines harmonized across cohorts. We conducted subgroup analyses and probed different operationalizations and subtypes of maltreatment, as well as interactions with age. Additional analyses stratified for sex were run for all models to account for potential sex-specific neural correlates of CM. Replicability was assessed by the spatial overlap of significant findings between our three cohorts, in addition to analyzing all cohorts together in a pooled model. We tested the hypothesis that CM is associated with lower gray matter volume (GMV).

Here, we show that there is little evidence for the replicability of gray matter correlates of childhood maltreatment across well-powered adult cohorts, using retrospective self-report measures. This is shown for VBM and for regional parcellation-based measures of cortical thickness and surface, as well as subcortical volume. Consistent non-replicability is presented across all maltreatment operationalizations (including CM subtypes and severe forms of CM), subgroup analyses (including individuals with or without MDD, or medication-naïve MDD patients) and in additional analyses stratified by sex. Similar non-replicability is observed for CM interactions with age. The largest evidence for maltreatment-associated gray matter effects is found in VBM analyses when not adequately controlling for confounding MDD diagnosis. In contrast, the association between childhood maltreatment and depression is found across a variety of different clinical characteristics and replicates consistently across all three cohorts.

## Results

### Associations of childhood maltreatment with demographic and clinical characteristics

CTQ scales were highly interrelated with each other, and they showed a pattern of small positive associations with age and small negative associations with education years (Fig. 1a). Furthermore, within the MDD participants CTQ scales showed a pattern of weak to moderate associations with previous and current clinical characteristics (Fig. 1a). Overall, the relationship between CM reports and demographic and clinical variables was highly similar across the three cohorts, except that age and number of inpatient treatments were not consistently associated with CTQ scales within the BiDirect cohort (Supplementary Figs. S2–S4). Participants with a MDD diagnosis reported significantly more severe CM, as compared to HC participants (Fig. 1b and Supplementary Table S5). This was found across all CM subtypes and highly consistent across all cohorts (Supplementary Fig. S5). Largest differences were found for the emotional abuse and neglect subscales (up to $r_{rank-biserial} = .517$).

### Voxel-based gray matter associations with childhood maltreatment – pooled sample across all cohorts

A total of 18 different statistical models were conducted for all brain-wide analyses. All conducted models are described in Table 1. Results using the full sample from pooling all cohorts together are presented at a conservative significance threshold of $p_{FWE} < .05$, corrected at the voxel-level. Findings from the pooled analyses are shown in Table 2 and Fig. 2.

When controlling for MDD diagnosis (Model 1), no voxels with a significant CM association were found. Dropping MDD diagnosis as covariate (Model 2) yielded significant widespread clusters (total $k = 5122$), located mainly within superior and middle temporal areas, a bilateral fusiform and lingual complex, the thalamus, as well as in the

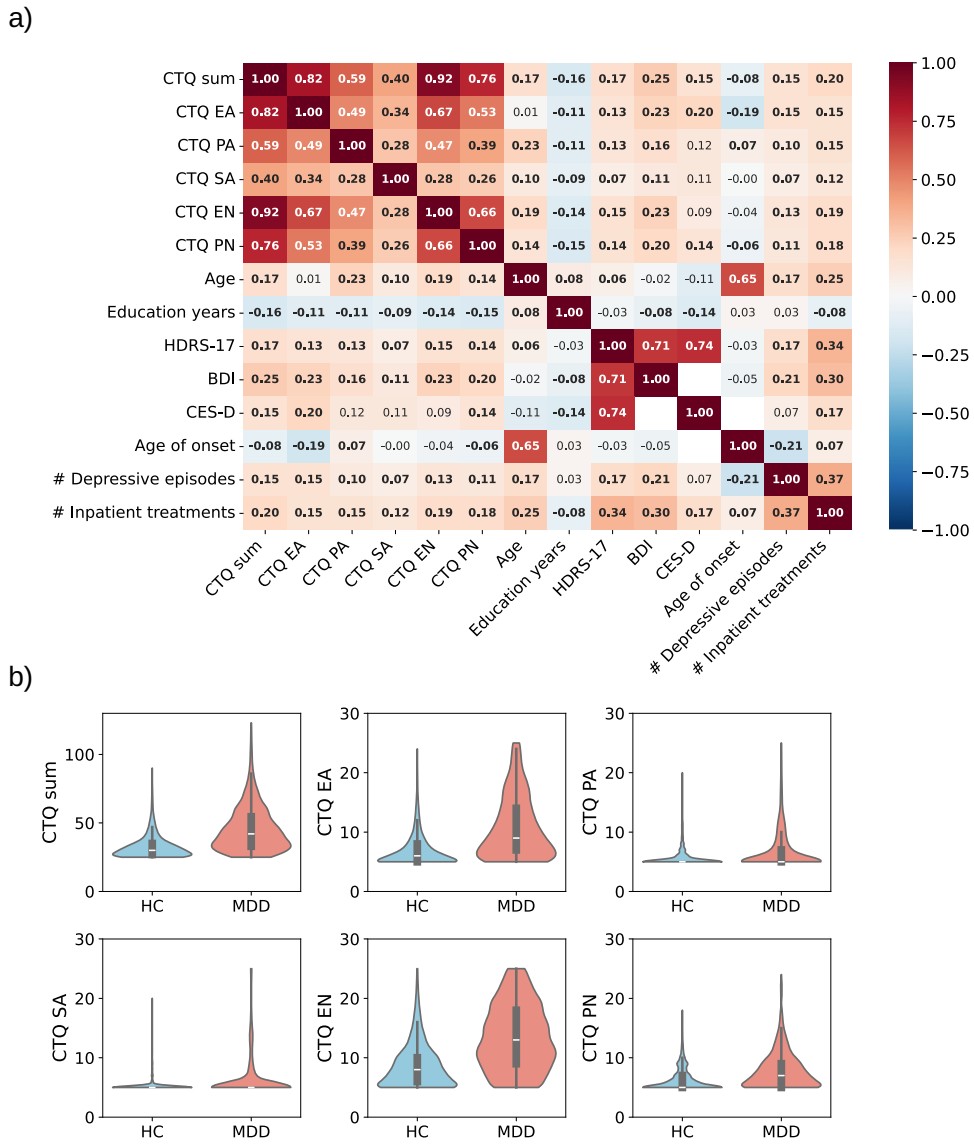

**Fig. 1 | Associations between CTQ scales, demographic variables and clinical variables.** *Note.* **a** Spearman correlations are shown (two-sided). All correlations involving clinical variables (HDRS-17, BDI, Age of onset, number of depressive episodes and number of inpatient treatments) were only calculated within the MDD subsample. The BDI was only available within MACS and MNC, while the CES-D was only available for the BiDirect cohort. Significant associations at $p < .05$ are shown in bold font. **b** Violin plots are shown depicting the distribution of the CTQ sum scale, as well as the five CTQ subscales (based on total $N = 3225$; HC $n = 1898$; MDD $n = 1327$). Boxplots within the violins depict the median (center line), the interquartile range (IQR; box from the 25th to 75th percentile), and whiskers extending to the most extreme values within 1.5 × IQR. CTQ, childhood trauma questionnaire; EA, emotional abuse; PA, physical abuse; SA, sexual abuse; EN, emotional neglect; PN, physical neglect; HDRS-17, 17-item Hamilton Depression Rating Scale; BDI, Beck Depression Inventory; CES-D, Center for Epidemiologic Studies Depression Scale; HC, healthy control; MDD, major depressive disorder.

orbitofrontal cortex and the insula. Subgroup analyses revealed small, significant clusters in HC individuals when using CTQ sum as a predictor (Model 3; total $k = 122$) within the medial orbitofrontal cortex, while no clusters survived the FWE-correction within the MDD sample (Model 4). Regarding subtypes of CM, no CTQ subscales were associated with GMV surpassing an FWE-corrected threshold, except a small cluster emerging when using physical abuse as a predictor (Model 8; total $k = 3$ within the thalamus). Similar results were obtained when investigating individuals with 'severe' maltreatment: again, the model without controlling for MDD diagnosis yielded widespread reductions in the group with severe maltreatment as compared to the group with 'none to minimal' maltreatment in widespread clusters (Model 13; total $k = 11256$). This effect was also found in much smaller localized clusters within HC samples only (Model 14; total $k = 140$). Effect sizes across models when pooling cohorts ranged between

partial $R^2 = .006$ and partial $R^2 = .022$. Interaction models yielded only small, significant clusters within HC samples only (Model 17; total $k = 2$).

Pooled analyses stratified by sex yielded similar results, however, with some additional clusters emerging in female subsamples when investigating severe CM in HC and MDD samples, while controlling for diagnosis (Model 12; total $k = 1847$). Overall, there was a descriptive pattern of more models yielding significant effects (and larger clusters) in the female subsamples as compared to the male subsamples. Results stratified by sex are shown in Supplementary Table S6 and S7.

## Replicability of voxel-based gray matter associations with childhood maltreatment across single cohorts

The same 18 models conducted in the pooled cohorts were also fitted in each cohort separately, using liberal uncorrected significance

**Table 1 | Summary of all conducted statistical models and the respective included sample sizes**

| Model | CM operalization | Controlled for MDD diagnosis | Subsamples | Sample size - n | | | |
|-------|------------------|------------------------------|------------|------|-----|---------|--------|
| | | | | MACS | MNC | BiDirect | Pooled |
| Model 1 | CTQ sum | yes | HC/MDD | 1752 | 916 | 557 | 3225 |
| Model 2 | CTQ sum | no | HC/MDD | 1752 | 916 | 557 | 3225 |
| Model 3 | CTQ sum | – | HC | 930 | 647 | 321 | 1898 |
| Model 4 | CTQ sum | – | MDD | 822 | 269 | 236 | 1327 |
| Model 5 | Abuse/threat | yes | HC/MDD | 1752 | 916 | 557 | 3225 |
| Model 6 | Neglect/deprivation | yes | HC/MDD | 1752 | 916 | 557 | 3225 |
| Model 7 | EA subscale sum | yes | HC/MDD | 1752 | 916 | 557 | 3225 |
| Model 8 | PA subscale sum | yes | HC/MDD | 1752 | 916 | 557 | 3225 |
| Model 9 | SA subscale sum | yes | HC/MDD | 1752 | 916 | 557 | 3225 |
| Model 10 | EN subscale sum | yes | HC/MDD | 1752 | 916 | 557 | 3225 |
| Model 11 | PN subscale sum | yes | HC/MDD | 1752 | 916 | 557 | 3225 |
| Model 12 | Extreme groups (none/severe) | yes | HC/MDD | None: 644 Severe: 348 | None: 369 Severe: 145 | None: 213 Severe: 98 | None: 1226 Severe: 591 |
| Model 13 | Extreme groups (none/severe) | no | HC/MDD | None: 644 Severe: 348 | None: 369 Severe: 145 | None: 213 Severe: 98 | None: 1226 Severe: 591 |
| Model 14 | Extreme groups (none/severe) | – | HC | None: 500 Severe: 51 | None: 324 Severe: 41 | None: 165 Severe: 17 | None: 989 Severe: 109 |
| Model 15 | Extreme groups (none/severe) | – | MDD | None: 144 Severe: 297 | None: 45 Severe: 104 | None: 48 Severe: 81 | None: 237 Severe: 482 |
| Model 16 | CTQ sum | – | MDD med-naïve | 334 | 39 | 34 | 407 |
| Model 17 | CTQ sum * age | – | HC | 930 | 647 | 321 | 1898 |
| Model 18 | CTQ sum * age | – | MDD | 822 | 269 | 236 | 1327 |

*Note.* In all models, we additionally controlled for age, sex and total intracranial volume. The extreme group comparisons are based on cutoff-based categorizations of severity. For models with comparison of extreme groups (Models 12-15), sample sizes are given for both the group with 'none to minimal' maltreatment (labeled 'none') and the group with 'severe' maltreatment. Models 17-18 contain an interaction term between CTQ sum and age as the predictor. CM, childhood maltreatment; CTQ, Childhood Trauma Questionnaire; HC, healthy controls; MDD, major depressive disorder; MACS, Marburg Münster Affective Disorders Cohort Study; MNC, Münster Neuroimaging Cohort; BiDirect, BiDirect study cohort; EA, emotional abuse; PA, physical abuse; SA, sexual abuse; PN, physical neglect; EN, emotional neglect.

**Table 2 | Results summary for pooled cohorts (*n* = 3225) at a significance level of $p_{FWE} < .05$**

| Model | k significant | partial $R^2$ | | |
|-------|---------------|-----|-----|-----|
| | | min | max | main regions |
| Model 1 | 0 | – | – | – |
| Model 2 | 5122 | 0.006 | 0.011 | Temporal Mid/Sup R, Fusiform L + R, Rectus L + R, Insula R, Lingual L, Parahippocampal R, Thalamus L |
| Model 3 | 122 | 0.010 | 0.012 | OFC Med L, Rectus L |
| Model 4 | 0 | – | – | – |
| Model 5 | 0 | – | – | – |
| Model 6 | 0 | – | – | – |
| Model 7 | 0 | – | – | – |
| Model 8 | 3 | 0.006 | 0.006 | Thalamus R[a] |
| Model 9 | 0 | – | – | – |
| Model 10 | 0 | – | – | – |
| Model 11 | 0 | – | – | – |
| Model 12 | 0 | – | – | – |
| Model 13 | 11256 | 0.011 | 0.021 | Temporal Mid/Sup R, Rectus L + R, Fusiform L + R, Cerebellum L + R, Insula R, Lingual L + R |
| Model 14 | 140 | 0.017 | 0.022 | Cerebellum R, Thalamus R[a] |
| Model 15 | 0 | – | – | – |
| Model 16 | 0 | – | – | – |
| Model 17 | 2 | 0.011 | 0.011 | OFC R[b] |
| Model 18 | 0 | – | – | – |

*Note.* The number of significant voxels is shown, as well as their minimum and maximum effect size for each analysis. Cluster labeling was conducted based on the aal atlas using the atlasreader Python package[102]. Main regions are reported. Results were derived from general linear models with a one-sided test of the respective maltreatment predictor, at $p_{FWE} < .05$. [a]Cluster labeling using aal resulted in 'no_label' however, checking Desikan-Killiany and Harvard-Oxford atlases indicated localization within the thalamus. [b]Cluster labeling using aal resulted in 'no_label' however, checking Desikan-Killiany and Harvard-Oxford atlases indicated localization within the orbitofrontal cortex. L, left; R, right; Mid, middle; Sup, superior; OFC, orbitofrontal cortex; med, medial.

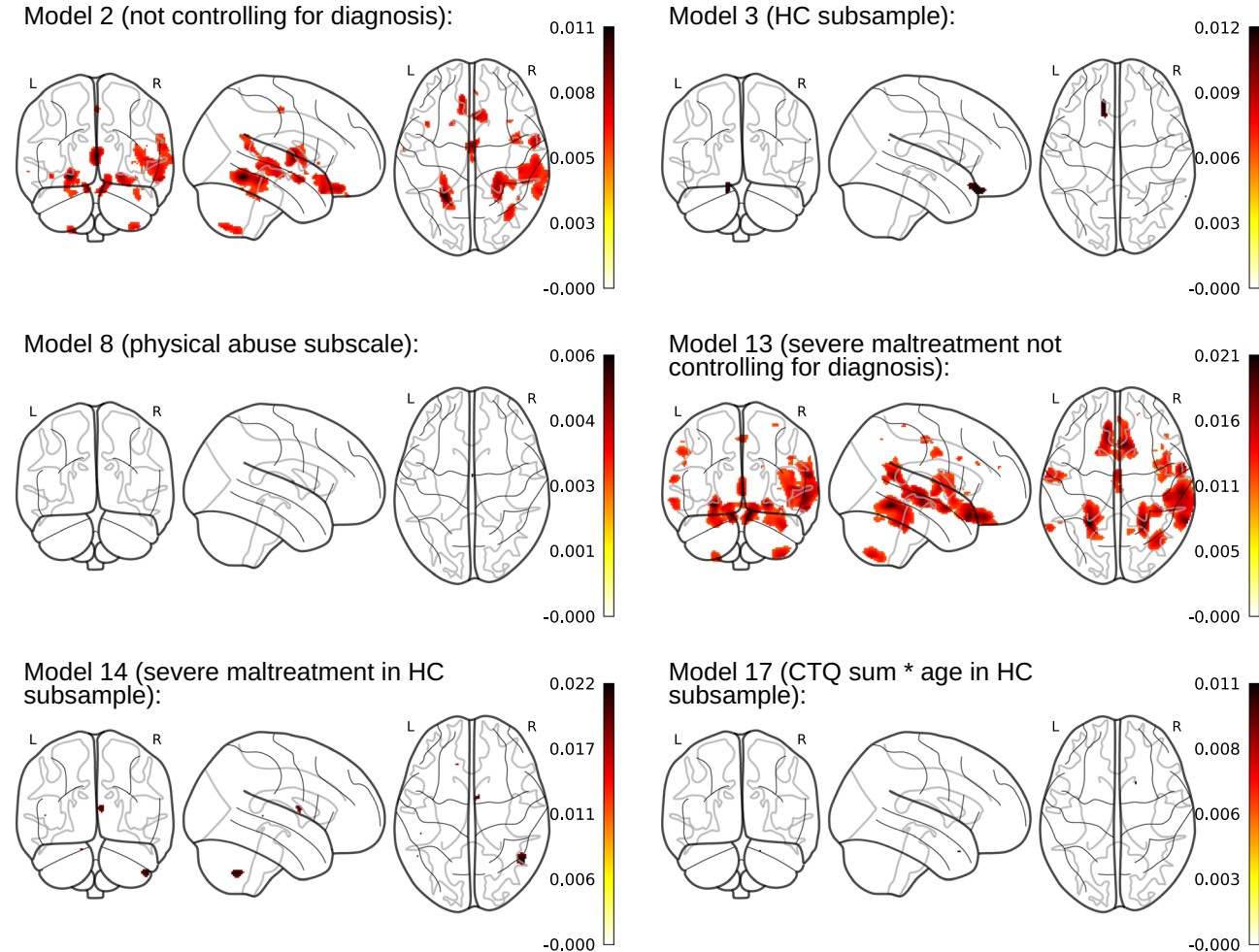

**Fig. 2 | Significant clusters from pooled analysis at $p_{FWE} < .05$.** *Note.* Glass brains are shown with maximum intensity projections. Color bars represent the partial $R^2$ of the CM predictor in the respective model (in Model 17 of the interaction term). Results were derived from general linear models with a one-sided test of the respective maltreatment predictor and a two-sided test of the interaction terms, both at $p_{FWE} < .05$. HC, healthy controls; MDD, major depressive disorder.

thresholds of $p_{unc} < .001$ and $p_{unc} < .01$. Within single cohorts, both significance thresholds and each statistical model yielded significant voxels in at least one of the three cohorts. In turn, each of the three cohorts produced significant voxels in most of the statistical models. The highest number of significant voxels was observed in Model 2 and Model 13 - both models where HC and MDD samples were included, but diagnosis was not included as a covariate. A detailed summary of cohort-wise results across models, including additional analyses stratified by sex, is shown in supplementary Tables S8–S13. Across probed models and across single cohorts, the nominally significant voxels were widespread throughout the brain, including the cerebellum, temporal and frontal areas, subcortical areas and somatosensory cortices.

Investigating replicability revealed that there was not one voxel that was congruently significant (i.e., replicable) at a threshold of $p_{unc} < .001$ in all three cohorts (Table 3). This finding was consistent across all probed statistical models, including HC and MDD subgroup analyses, testing subtypes of CM and comparing groups with severe CM and no CM, as well as when testing age interactions. Similarly, comparing pairs of cohorts also yielded no voxels that regionally overlapped between any pairwise cohort combinations, for most of the tested models. Only two models yielded marginal pairwise overlap in significance at this threshold: when testing the physical neglect subscale of the CTQ (Model 11), there was a small overlap between the MNC and BiDirect cohorts located within the supramarginal gyrus

(overlap $k = 3$; Dice =.002). Furthermore, there was an overlap of $k = 2$ voxels (Dice = .001) between the MACS and the BiDirect cohort in Model 13 (comparing extreme groups without controlling for diagnosis). This extent of replicability was not significant ($p_{FDR} > .611$), as indicated by permutation-based null-distributions of overlap across cohort-combinations.

When rerunning the replicability analyses using an even more liberal threshold of $p_{unc} < .01$ the observed spatial overlap in significant voxels was increased across models. The only models yielding any overlap across all three cohorts at this threshold were Model 2 (CTQ sum; not controlling for MDD diagnosis), with converging significance in $k = 4$ voxels, and Model 13 ($k = 12$ voxels; comparing extreme groups of CM, not controlling for MDD diagnosis). Pairwise cohort combinations yielded additional spatial overlap in significance across models, with maximum overlap in Model 13 ($k = 1329$, Dice = 0.081). All observed overlap of any cohort combination was non-significant. This was consistent across all models (all $p_{FDR} > .144$), as indicated by permutation-based null-distributions of overlap across cohort-combinations.

Replicability results were largely consistent when rerunning all analyses stratified by sex. A summary of the extent of spatial overlap of effects across cohorts, as well as the significance of this replicability, is shown in Table 3 and Supplementary Tables S14–S18. Significant clusters across significance thresholds, cohorts and statistical models are shown in Fig. 3 and Supplementary

**Table 3 | Replicability across cohorts indicated by spatial overlap in significance at a level of p_unc < .001**

| | MACS-MNC | | MACS-BiDirect | | MNC-BiDirect | | MACS-MNC-BiDirect | |
|---|---|---|---|---|---|---|---|---|
| | k overlap | DICE | k overlap | DICE | k overlap | DICE | k overlap | DICE |
| Model 1 | 0 | 0 | 0 | 0 | 0 | 0 | 0 | – |
| Model 2 | 0 | 0 | 0 | 0 | 0 | 0 | 0 | – |
| Model 3 | 0 | 0 | 0 | 0 | 0 | 0 | 0 | – |
| Model 4 | 0 | 0 | 0 | 0 | 0 | 0 | 0 | – |
| Model 5 | 0 | 0 | 0 | 0 | 0 | 0 | 0 | – |
| Model 6 | 0 | 0 | 0 | 0 | 0 | 0 | 0 | – |
| Model 7 | 0 | 0 | 0 | 0 | 0 | 0 | 0 | – |
| Model 8 | 0 | 0 | 0 | 0 | 0 | 0 | 0 | – |
| Model 9 | 0 | 0 | 0 | 0 | 0 | 0 | 0 | – |
| Model 10 | 0 | 0 | 0 | 0 | 0 | 0 | 0 | – |
| Model 11 | 0 | 0 | 0 | 0 | 3[a] | 0.002 | 0 | – |
| Model 12 | 0 | 0 | 0 | 0 | 0 | 0 | 0 | – |
| Model 13 | 0 | 0 | 2[b] | 0.001 | 0 | 0 | 0 | – |
| Model 14 | 0 | 0 | 0 | 0 | 0 | 0 | 0 | – |
| Model 15 | 0 | 0 | 0 | 0 | 0 | 0 | 0 | – |
| Model 16 | 0 | 0 | 0 | 0 | 0 | 0 | 0 | – |
| Model 17 | 0 | 0 | 0 | 0 | 0 | 0 | 0 | – |
| Model 18 | 0 | 0 | 0 | 0 | 0 | 0 | 0 | – |

*Note.* The overlap in significant voxels is presented across all statistical models and all cohort combinations. The DICE score is presented for pairwise combinations with any voxels overlapping. Results were derived from general linear models with a one-sided test of the respective maltreatment predictor. Overlap is calculated based on significant clusters at uncorrected $p < .001$ per cohort-wise analysis. MACS, Marburg Münster Affective Disorders Cohort Study; MNC, Münster Neuroimaging Cohort; BiDirect, BiDirect study cohort. [a]Corresponding permutation-based significance of overlap: $p = .011$, $p_{FDR} = .611$. [b]Corresponding permutation-based significance of overlap: $p = .017$, $p_{FDR} = .611$.

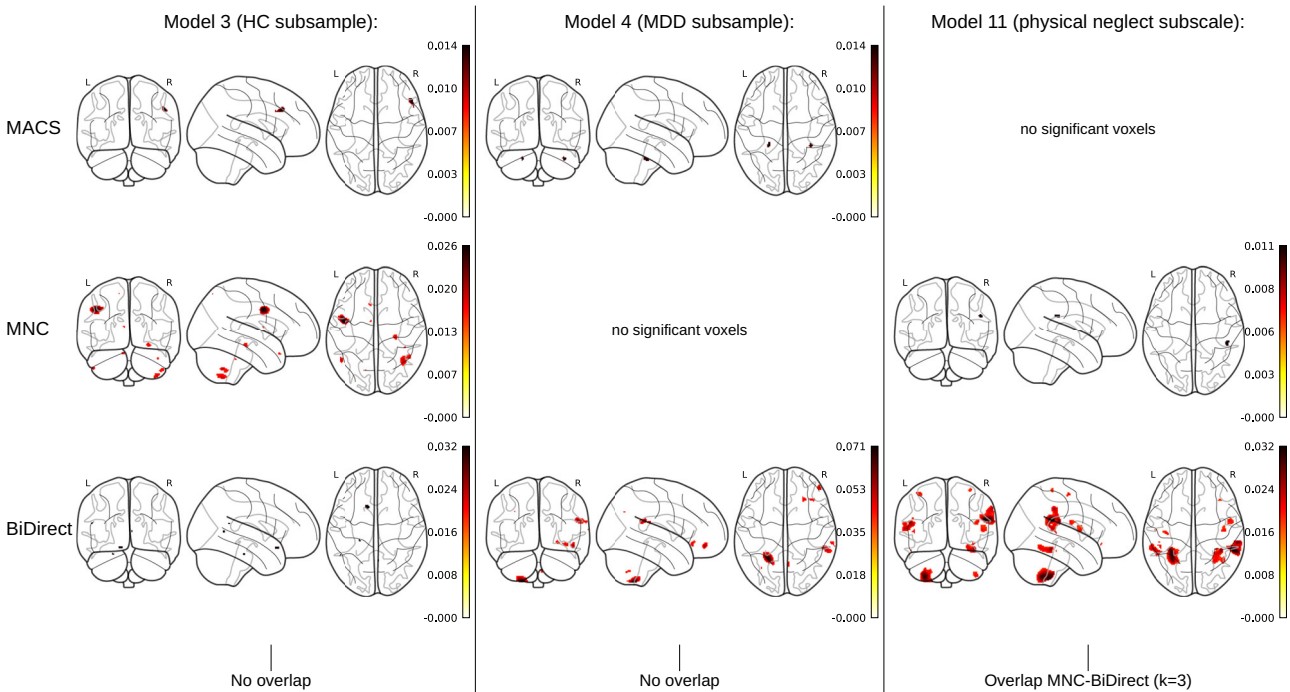

**Fig. 3 | Significant clusters across cohort-wise analyses at p_unc < .001 for exemplary models (replicability analysis).** *Note.* Glass brains are shown with maximum intensity projections. Color bars represent the partial $R^2$ of the maltreatment predictor in the respective model. Exemplary model results are shown for two models without any spatial overlap (i.e., replicability), namely the CTQ sum analysis in HC participants (Model 3) and in MDD participants (Model 4), as well as one model showing some degree of spatial overlap (Model 11 testing the physical neglect subscale as a predictor). Results were derived from general linear models with a one-sided test of the respective maltreatment predictor, at uncorrected $p < .001$. HC, healthy controls; MDD, Major Depressive Disorder; MACS, Marburg Münster Affective Disorders Cohort Study; MNC, Münster Neuroimaging Cohort; BiDirect, BiDirect cohort.

Figs. S6–S23, with additional results stratified by sex presented in Supplementary Figs. S24–S59.

## Parcellation-based and global gray matter density associations with childhood maltreatment

Investigating parcellation-based regional measures of cortical thickness and surface, as well as subcortical volume, yielded similar results as voxel-based analysis. For the pooled analysis, the largest significant effect was found in Model 12 for the surface of the right banks of the superior temporal sulcus (partial $R^2$ = .007). Global gray matter density showed a significant negative association with CM in several models when using the pooled cohorts (with a maximum effect size in pooled Model 13; partial $R^2$ = .017). Parcellation-based and global gray matter density results significant at $p_{FDR} < .05$ are shown in Supplementary Table S19. Replicability analysis revealed that only Model 7 (CTQ EA subscale) showed minor overlap in significance in one region (thickness of the left precuneus), in female participants only. All other combinations of cohorts, models, and sex-stratification did not yield any overlap in regional significance for any cortical or subcortical measure, including aggregate measures of cortical thickness and cortical surface for the left and right hemisphere. The association between global gray matter and CM was also not replicable across cohorts in any model. Replicability results are shown in Supplementary Table S20.

## Discussion

Findings implicating long-term effects of CM on the structural morphology of the brain have been frequently published over the last decades and are central to a neurobiological model of how environmental risk is conveyed to psychopathology. However, in an unprecedented replication effort, we present evidence that localized brain-wide associations between gray matter structure and various operationalizations of CM are essentially non-replicable. This lack of replicability was consistently shown for a wide variety of common statistical approaches, across non-clinical and clinical, as well as sex-stratified subgroups and across a variety of operationalizations of CM, and for interactions with age. Consistent non-replicability was furthermore observed using different complementary methodologies to assess gray matter structure. Central limitations arise due to the concrete assessment method of CM utilized in this study (retrospectively via the CTQ) and due to low demographic (particularly ethnic) variability of the included samples. The extensive non-replicability of CM-related gray matter effects were contrasted by highly replicable negative associations of CM with MDD diagnosis and with various measures of current and previous depression severity.

In our pooled analysis, we found small, significant effects within HC subsamples when investigating physical abuse. Furthermore, in the pooled analysis, we identified large and widespread clusters when not controlling for confounding MDD diagnosis. Notably, the localized clusters seemingly associated with CM in this latter analysis largely overlap with clusters that were identified to be associated with a lifetime MDD diagnosis in a systematic case-control study using the same cohorts[30].

When using liberal uncorrected thresholds, we found a vast array of regions seemingly associated with CM across the different cohorts and statistical models. In isolation, each of these results could easily have been the basis of a publication, just like several smaller existing studies in the field, including previous publications from our own group[4,10]. Importantly, these widespread significant effects for each single cohort should not be interpreted as solid evidence for effects due to the liberal significance thresholds and resulting massively inflated alpha error (i.e., false-positives). In our most liberal analyses, some extent of replicability was observed, particularly when investigating HC and MDD samples together, while not controlling for MDD

diagnosis. However, the identified overlap was small even for pairwise combinations of cohorts. Furthermore, permutation-based significance testing of this descriptive overlap indicated that it was not higher than expected by chance. Overall, our findings suggest that gray matter reductions associated with CM are non-replicable. Similar results were obtained when rerunning all analyses stratified by sex and for testing age interactions with CM. Even highly cited previous reports of brain structural correlates of CM, such as associations with lower GMV within the hippocampus[4,10] could not be confirmed.

Null findings are always difficult to interpret due to a multitude of potential reasons for failing to detect an effect. Potential reasons for false-negatives can stem from the specific measurement and operationalization of the predictor or dependent variable, the specific statistical approach (e.g., inclusion of covariates), insufficient statistical power, the sample selection and, regarding replicability, differences between specific cohort characteristics. In the following we will discuss each of these potential sources of effect variability.

Despite of its common use, the CTQ has been criticized for neglecting the timing of maltreatment[31] and showing low agreement with prospective CM measures[32,33], the latter likely due to memory and reporting biases[34]. Although depressive states are thought to bias retrospective reports of CM, we found CTQ scores to be highly stable over two years within the MACS and MNC cohorts, with no systematic association with changes in depressive symptoms[35]. Still, memory or reporting biases cannot be ruled out completely. Regardless, retrospective CM measures generally show stronger associations with later psychopathology than prospective measures, potentially measuring differing entities[36]. In addition, timing of exposure may critically moderate CM's effects on clinical[37–39] and neurobiological endpoints[40–42]. Thus, it is unclear whether our findings, based on the retrospective CTQ, generalize to prospectively assessed CM measures or those incorporating timing information. While the general critique of the CTQ may be valid and posits an important limitation to the current findings, it is notable that the CTQ is also the instrument which has been used in most of the referenced studies, which have reported GMV associations with CM (e.g., Dannlowski et al.[4], Gold et al.[43], Opel et al.[10], Paquola et al.[44], Samplin et al.[45], Whittle et al.[46]), ensuring methodological comparability with our study. Some of the studies reporting on significant effects use alternative assessment instruments to measure CM, such as interview measures[43], or CTQ data from adolescent or young adult samples, potentially less affected by memory biases[43,44,46]. Furthermore, our analyses are cross-sectional, and while longitudinal studies are rare, some have shown how CM may affect brain development and links to psychopathology over time[7,44,47,48]. Future studies should expand our current findings and investigate the replicability of CM neural correlates using alternative instruments, designs and samples. Long-term longitudinal studies from childhood to adulthood are necessary to robustly identify causal pathways between CM, neurobiological alterations and clinical consequences.

On the side of the dependent variable, we used state-of-the-art voxel-wise GMV assessments. Meta-research on neuroimaging replicability suggests that the researchers' degrees of freedom regarding scanning parameters, preprocessing and quality control pipelines contribute to low reproducibility and replicability[49,50]. While such differences could influence findings, our procedures were closely harmonized across cohorts, making this an unlikely explanation for low replicability in our study. In addition, we showed that our results were consistent when using a complementary parcellation-based approach, operationalizing gray matter structure as regional cortical thickness and surface, as well as subcortical volume. However, the possibility remains that a change in preprocessing parameters may lead to different results. Furthermore, it remains unclear whether our null

findings generalize to other imaging modalities, such as functional or structural connectivity.

Similarly, several decisions are required regarding the operationalization of predictors and statistical modeling. In previous studies, CM (even when based on the CTQ) was operationalized in different ways (total sum score, subscale scores, different cutoff-based categories), and confounding psychopathology was differently addressed. Importantly, investigating CM effects in clinical samples always comes with the challenge that CM and psychopathology (diagnosis, as well as symptom severity) are inherently highly confounded. This is particularly relevant for studies including HC and MDD samples together, rendering a disentanglement of CM effects and psychopathology effects virtually impossible, limiting the interpretability of findings. This problem is increased when samples are not drawn from a community sample, thus oversampling MDD patients and consequently overestimating the confounding between CM and MDD. While controlling for diagnosis potentially leads to an underestimation of CM effects (e.g., Chaney et al.[51], S. Lu et al.[52], X. W. Lu et al.[53]), not controlling for diagnosis could overestimate CM effects due to underlying neural correlates of diagnosis. Notably, a null finding when controlling for psychopathology could even be consistent with theories of how the clinical consequences of CM are neurobiologically conveyed due to inherently assumed shared variance. To achieve good comparability with previous studies, we employed a comprehensive approach that included a range of common statistical models, including investigations of clinical and non-clinical populations separately. While our list of models is not exhaustive and our conclusions are limited to these specific approaches, the consistent finding of poor replicability across all tested models is striking.

Insufficient statistical power may partly account for low replicability, especially for some subgroup analyses. Although our study represents the largest replicability investigation of brain alterations associated with CM to date, it may still lack power to detect small effects typical in biological psychology and psychiatry, which rarely exceed 2.5% explained variance[28,54]. This limitation is particularly relevant for some subgroup analyses, affecting particularly the interpretability of replicability findings in medication-naïve patient samples due to the scarcity of these subsamples in two of the three cohorts. Previous meta-analytic evidence, however, does not indicate that additional effects show in medication-naïve samples that are masked in medicated patient samples[14]. Comparably small samples were reached for the extreme CM severity groups within HC samples, again limiting the interpretability of observed non-replicability. Similarly, power problems can arise due to insufficient variability of a predictor in a sample, which could be the case in HC samples, which showed considerably lower mean CM severity and lower variability in CM reports. However, our distribution of reported CM severity in HC samples is comparable with previous studies reporting on significant gray matter effects of CM in non-clinical samples, while our samples in each cohort are large compared to previous studies (e.g., Dannlowski et al.[4]). For most analyses, our attained sample size and statistical power are well within the realm of previous meta-analyses[13,14] and large-scale consortia analyses[18,55].

Sample selection is a key source of variability. While our cohorts encompass a broad range of maltreatment severity and clinical characteristics, they are relatively homogeneous demographically, consisting of German individuals of Western European ancestry with relatively high education levels. This homogeneity enhances conditions for replicability but limits the generalizability of our null findings to other populations. Differences between cohorts may also contribute to low replicability. For instance, the BiDirect cohort was considerably older, while the MNC cohort included only acutely depressed inpatient MDD patients, compared to a mix of outpatient and remitted individuals in the other cohorts. Differences in current depression

severity and illness history were also observed across MDD samples. However, the extent of non-replicability across all pairwise cohort comparisons suggests that cohort-specific differences alone are unlikely to fully explain our null findings.

All these aspects pose potential sources of effect variability and could account for false-negative findings. However, previous studies reporting CM associations were highly comparable regarding the utilized methodology. It should be noted that it is possible that conventional conceptualizations of clearly localizable gray matter reductions due to CM on a group level could be too simplistic. Recently, machine learning and normative modeling approaches have been increasingly promoted following the notion that the concrete shape of neurobiological consequences in the brain may be highly individual[56–58].

The absence of evidence cannot directly be interpreted as evidence for the absence of a phenomenon[59]. However, the extent of non-replicability of gray matter correlates of CM still appears disconcerting. Notably, this is in contrast to the replicability of GMV reductions linked to lifetime MDD, observed using a similar approach[30]. Replicability is a fundamental principle of the scientific process and essential for accumulating scientific knowledge[60]. However, non-replicability of published findings is a growing concern across disciplines, such as cell biology[61], genetics[62,63], oncology[64], epidemiology[65] and psychology[66]. Factors contributing to this *replication crisis* include publication pressure, bias toward positive results, and analytic flexibility[67,68]. Neuroimaging, with its high analytic flexibility, numerous tests, and small, underpowered samples, is particularly susceptible to overestimated effect sizes and non-replicability[69,70]. Recent research shows that thousands of participants may be needed for robust, replicable brain-wide associations due to small true effect sizes[28], which has sparked ongoing debates in the field (e.g., Bandettini et al.[71], Genon et al.[72], Nour et al.[73], Rosenberg & Finn[74], Spisak et al.[75], Tervo-Clemmens et al.[76]). Our study supports this view, suggesting that low replicability may be a broader issue, not limited to this research question alone. Notably, no consensus exists on how to define "successful" replicability in voxel-based neuroimaging. We contribute to this by formalizing and testing cross-sample replicability of voxel-based analyses.

Concerns about low replicability have led to the development of open science policies, such as preregistration of hypotheses and analysis plans, as well as comprehensive disclosure of analysis code and results[77], to achieve transparency and reduce biases. Accordingly, scholars have increasingly advocated for these practices to enhance replicability in neuroimaging research[78]. However, replications and open science practices remain very rare in neuroimaging[79]. In addition, approaches like cross-validation, which assess the generalizability of statistical findings to independent data, can help identify overestimated effect sizes and non-replicability in smaller samples[80].

Our findings underline the importance of taking a step back and shifting the focus towards increasing and investigating the replicability and generalizability of presumably established research findings. Various open science practices are available for this: (1) preregistrations of hypotheses and analyses, (2) transparent sharing of analysis code and methods, and the publication of comprehensive (i.e., non-thresholded) results[81], as well as (3) the mere execution of direct and conceptual replication studies, and (4) the publication of null findings. Open science practices should be routinely adopted in neuroimaging research on mental disorders to increase replicability and thus maximize the potential for clinical translation.

## Methods
### Participants
Samples from three large-scale independent cohorts were included in the present analysis: the Marburg Münster Affective Disorders Cohort Study (MACS), the Münster Neuroimaging cohort (MNC) and the BiDirect cohort. All three cohorts include adults (age 18–65 years) with

and without mental disorders. All research procedures complied with the Declaration of Helsinki and relevant ethical regulations, and were approved by the respective local ethics committees at the University of Münster (MACS: AZ 2014-422-b-S; MNC: AZ 2007-307-f-S; BiDirect: AZ 2009-391-f-S) and by the Medical Faculty of the University of Marburg (MACS: 07/14). All participants provided written informed consent.

Recruitment was restricted to individuals proficient in the German language and with Western European ancestry (as the cohorts were originally conceptualized for genetic analyses). For the current analyses, we included healthy control (HC) individuals, as well as individuals with a lifetime MDD diagnosis. In total, a sum of $n = 3225$ participants were included (HC: $n = 1898$; MDD: 1327). Identical exclusion criteria were applied for all three independent cohorts: (1) duplicate cases resulting from individuals that were included in more than one of the utilized cohorts, (2) presence of a lifetime bipolar disorder, psychosis spectrum disorder or substance dependencies (other psychiatric comorbidities were permitted), (3) severe head trauma or severe/chronic somatic illness (e.g., Parkinson's disease, multiple sclerosis, stroke, myocardial infarction), (4) missing MRI data and image artefacts diagnosed during quality control, (5) missing data in the CTQ.

For details on the methods and general inclusion criteria of the study samples, we refer to previous publications (MACS: Kircher et al.[82], Vogelbacher et al.[83], MNC: Dannlowski et al.[84], Opel, Redlich, Dohm, et al., 2019; BiDirect: Teismann et al.[85]) and to the supplements. The final samples included in the current analyses comprised $n = 1752$ participants from MACS (HC: $n = 930$; MDD: $n = 822$), $n = 916$ participants from MNC (HC: $n = 647$; MDD: $n = 269$), and 557 participants from BiDirect (HC: $n = 321$; MDD: $n = 236$). Across cohort samples, the mean age was 38.18 (SD = 13.65), with 60% reporting female sex ($n = 1934$). In the MACS cohort, the mean age was 35.30 years (SD = 13.09), with 64.7% female ($n = 1134$); in the MNC cohort, 35.44 years (SD = 12.53), with 55.4% female ($n = 507$); and in the BiDirect cohort, 51.77 years (SD = 7.66), with 52.6% female ($n = 293$).

Detailed sample characteristics of the three cohorts, including demographics, reports of CM and clinical characteristics, are described in Supplementary Tables S1 and S2. Differences between cohorts in sample characteristics are shown in Supplementary Table S3, while differences in clinical characteristics between the MDD subsamples of the cohorts are shown in Supplementary Table S4. Age distributions across cohorts and diagnosis groups are shown in Supplementary Fig. S1.

Of note, findings regarding gray matter correlates of CM have been previously published using MNC data at earlier stages of data assessments. However, these analyses only included a fraction of the data available for the current analysis (the largest sample including $n = 170$ subjects; Dannlowski et al.[4], Opel et al.[10]).

## Assessment of childhood maltreatment and clinical characterization

CM was assessed using the German version of the Childhood Trauma Questionnaire (CTQ)[86,87]. The CTQ is a 25-item retrospective self-report questionnaire capturing five different subtypes of CM, namely emotional abuse, physical abuse, sexual abuse, emotional neglect, and physical neglect. Each of these subtypes can be scored separately using a sum score of the five corresponding items, while the sum of these subscale scores amounts to the total CTQ score (expressing the total severity or load of experienced maltreatment). In addition, a categorical scoring of the CTQ has been introduced based on validated subscale cutoff values, dividing scores in each subscale into different severity categories from 'none to minimal' to 'severe'[88]. As described in detail below, here we utilize the total and subscale sum scores, as well as categorical cutoffs for group comparisons. The CTQ has been used in several hundreds of studies across different nationalities and validated on clinical and non-clinical populations[89]. It has been extensively

tested for its psychometric properties in several languages and geographic contexts[87,89–96].

Lifetime clinical diagnosis was assessed using structured clinical interviews by trained study personnel in each cohort. Within MACS and MNC, the German version of the Structured Clinical Interview for the DSM-IV (SCID; Wittchen et al., 1997) was used. Within the BiDirect cohort, the Mini International Neuropsychiatric Interview (MINI; Decubrier et al., 1997) was used, also based on the DSM-IV criteria. Further clinical characterization was done using a variety of standardized clinical interviews, rating scales and self-report questionnaires, capturing information on current remission status, current depression severity and previous course of disease (see supplements).

## Structural image acquisition and processing

T1-weighted high-resolution anatomical brain images were acquired using a 3 T MRI scanner using highly harmonized scanning protocols across all three cohorts. For the MACS sample, two different MRI scanners were used at the recruitment sites in Marburg (Tim Trio, Siemens, Erlangen, Germany; combined with a 12-channel head matrix Rx-coil) and Münster (Prisma, Siemens, Erlangen, Germany; combined with a 20-channel head matrix Rx-coil). MNC and BiDirect samples were both scanned using a Gyroscan Intera scanner with Achieva update (both by Philips Medical Systems, Best, The Netherlands).

Image preprocessing for VBM was conducted using the CAT12-toolbox (Gaser et al.[97], https://neuro-jena.github.io/cat/) using default parameters equally for all cohorts. Briefly, images were bias-corrected, tissue classified, and normalized to MNI-space using linear (12-parameter affine) and non-linear transformations, within a unified model including high-dimensional geodesic shooting normalization[98]. The modulated gray matter images were smoothed with a Gaussian kernel of 8 mm FWHM. Absolute threshold masking with a threshold value of 0.1 was used for all second-level analyses as recommended for VBM analyses (http://www.neuro.uni-jena.de/cat/). Image quality was assessed by visual inspection as well as by using the check for homogeneity function implemented in the CAT12 toolbox. Image acquisition and processing for all study samples were extensively described elsewhere[7,83,85]. Preprocessing steps for a complementary parcellation-based approach are described within the supplements.

Image harmonization was conducted using the Combat[99] with default parameters to control for differences in scanner hardware and corresponding effects on brain images. This procedure allows the specification of 'biological covariates' that are excluded from harmonization in order to preserve the desired variance of potentially confounding variables. We defined CTQ sum, age, sex, total intracranial volume (TIV) and MDD diagnosis as such covariates. The harmonization process was conducted across six different scanner groups: MACS Münster scanner, MACS Marburg scanner before and after body coil change, MNC scanner before and after gradient coil change and the BiDirect scanner setting.

## Statistical analysis

Associations between CM reports, demographic and clinical variables were investigated using Spearman correlations (due to highly non-normal distributions in the CTQ scales) and Mann-Whitney tests. For the latter, rank-biserial correlations were calculated as a measure of effect size.

Brain-wide associations between CM and voxel-wise GMV were tested using general linear models in a mass-univariate VBM approach. The available cohorts were investigated in two different steps: In a first step, we pooled all cohorts together in order to harvest the maximum sample size and thus the maximum available statistical power. For this pooled analysis, we used a voxel-wise family-wise error (FWE)-corrected significance threshold of $p_{FWE} < .05$.

In a second step we investigated the cross-cohort replicability by analyzing each of the cohorts separately using two liberal uncorrected significance thresholds of $p_{unc} < .001$ and $p_{unc} < .01$. Here, we examined the spatial convergence (i.e., overlap) of significant voxels across the cohorts as conjunctive criteria (convergence either across any subset of two cohorts or across all three cohorts). Note that these liberal significance thresholds should not be used by themselves for statistical inference due to a massively inflated alpha error from the mass-univariate testing. However, we defined these liberal thresholds as minimum thresholds for effects to be recognized as replicable. Importantly, the probability of finding the same voxel in two or even three cohorts constitutes a higher threshold than a voxel becoming significant just in a single cohort. In numbers, a threshold of $p < .001$ exceeded in each of the three single cohorts results in an effective false-positive rate of $p < .001^3 = .000000001$, while a threshold of $p < .01$ exceeded in any pair of two cohorts results in an effective false-positive rate of $p < .01^2 = .0001$. No extent threshold for minimum cluster size was used in any analysis. In order to test the significance of the replicability (i.e., overlap in effects) we applied permutation testing, permuting the respective predictor label for each cohort $k = 1000$ times subsequently obtaining a null distribution for the overlap analysis (how much overlap between cohorts can be expected by chance; exemplary permutations shown in Supplementary Fig. S60). Obtained p-values were FDR-corrected using the Benjamini-Hochberg procedure[100] across 18 models for which overlap was investigated across four different cohort-combinations (resulting in correction of sets of 72 tests).

The following statistical models were probed for (1) pooled analyses and (2) cross-cohort replicability analyses, to delineate the conditions under which gray matter associations may become evident (an overview of all statistical models is presented in Table 1). In all models, age, sex and TIV were included as covariates. In addition, lifetime MDD diagnosis was included as a control variable in all models unless stated otherwise:

In Model 1, we used the CTQ sum as the main predictor of interest, while additionally controlling for lifetime MDD diagnosis, as this variable is highly confounded with CM. Thus, in this model, we tested the effect of CM on GMV beyond any effect of diagnosis.

The model described above may not be sufficiently sensitive to detect CM effects due to substantial shared variance with the MDD diagnosis effect being partialized out. Therefore, we further tested a second model (Model 2), removing MDD diagnosis as a covariate. This allowed us to obtain a liberal estimate for the association between CM and gray matter, which, however, is not clearly separable from any MDD diagnosis effects.

To investigate CM effects independently from a confounding MDD diagnosis effect, we further conducted subgroup analyses within HC (Model 3) and MDD (Model 4) samples separately.

Based on the dimensional model of adversity[22,23] we probed associations between CM and GMV specifically for the abuse subscales (Model 5) and the neglect subscales (Model 6) of the CTQ, to differentiate between threat- and deprivation-related experiences.

In order to delineate effects of specific subtypes of CM in further detail, we tested a series of models with each of the five CTQ subscale sum scores as predictors respectively (Models 7–11).

Furthermore, we investigated the effects of severe forms of CM. This was done by identifying participants exceeding the subscale cutoff score for severe CM in any CTQ subscale, as defined and validated by Bernstein et al.[101]. This group of individuals with severe CM is contrasted with a control group that does not exceed any subscale cutoff (only CM reported within the range of 'none to minimal'). With this, we accounted for the notion that CM associations with VBM may become evident, particularly in individuals with severe experiences of CM. In our sample, $n = 591$ individuals (HC: $n = 109$; MDD: $n = 482$) fulfilled the criteria for severe maltreatment

in at least one of the CTQ subscales, while $n = 1226$ individuals (HC: $n = 989$; MDD: $n = 237$) fell into the category 'none to minimal' maltreatment experiences. The extreme group comparisons were done in the full sample (HC and MDD), in one model controlling for MDD diagnosis (Model 12) and in another model dropping MDD diagnosis as a covariate (Model 13). The distribution of severe CM across diagnostic groups was significantly uneven (Chi$^2$ = 645.71; $p < .001$, OR = 18.45, 95%-CI: [14.348, 23.732]). This strongly unequal distribution led us to conduct severity group analyses additionally in HC (Model 14) and MDD (Model 15) subgroups separately.

As an additional sensitivity analysis, we repeated the MDD subgroup analysis in patients without current antidepressant medication intake (Model 16).

Lastly, we investigated the potential moderating role of age for CM effects by including an additional interaction term between CTQ sum and age in the subsample Models 3 and 4, resulting in interaction Model 17 (in HC subsamples) and interaction Model 18 (in MDD subsamples).

To account for potential sex-specific effects, all analyses were additionally rerun stratified by female and male sex, as self-reported by the participants. One-sided negative contrasts were tested in all models probing main effects (i.e., CM associated with lower gray matter volume) due to poor evidence for potential positive associations between CM and VBM[6,13,14,16]. In contrast, two-sided tests were used for investigations of interaction effects (Model 17 and Model 18). Partial R$^2$ was calculated based on t-maps and reported as an effect size for the partialized percentage of explained variance of the respective CM predictor for each analysis.

Complementary to the voxel-wise approach, we reran all analyses described above using parcellation-based regional measures of cortical thickness and surface, as well as subcortical volume as dependent variables. Model specifications were identical to voxel-based analyses except that TIV was dropped as a covariate when investigating cortical thickness measures. In addition, global gray matter effects were investigated by calculating the average gray matter density across all voxels per individual (also tested without TIV as a covariate). Obtained p-values for regional parcellation-based measures and the global gray matter density variable were FDR-corrected using the Benjamini-Hochberg procedure[100] for each model separately, each time correcting for sets of 155 statistical tests (14 subcortical measures, 140 cortical measures and the global gray matter density variable).

All analyses were conducted using Python (version 3.9.12). Analyses were not preregistered.

**Reporting summary**

Further information on research design is available in the Nature Portfolio Reporting Summary linked to this article.

## Data availability

Source data for all Tables and Figures are provided as a Source Data file. For all figures containing brain plots, source data are provided as NIfTI files. These files contain the statistical values and thresholds underlying each brain visualization. Comprehensive non-thresholded statistical estimates for all analyses are made openly available via the OSF. Individual raw data is not published due to current EU data protection regulations and the sensitive nature of clinical MRI data but can be made available in form of summary statistics or anonymized aggregation of voxel-wise data upon reasonable request to the corresponding author, within four weeks, depending on the required data or results derivates. Source data are provided in this paper.

## Code availability

The code used for analysis is publicly available in an Open Science Framework (OSF) repository (https://osf.io/j8d9r/?view_only=

9edf436ab18f4e8db9ef4c71c4ac356c), to foster transparency and reproducibility of our analyses[29].

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

## Acknowledgements

We are deeply grateful to everyone who contributed to data collection and curation for the MACS, MNC, and BiDirect cohorts. This large-scale collaborative effort was made possible by the dedicated work of research and clinical staff, as well as student assistants and interns over many years. We are especially indebted to all study participants for their invaluable contributions. Funding: The MACS and MNC studies are funded by the Deutsche Forschungsgemeinschaft (DFG, German Research Foundation; grant FOR2107 DA1151/5-1, DA1151/5-2, DA1151/9-1, DA1151/10-1, DA1151/11-1 to U.D.; SFB/TRR 393, project grant no 521379614), the Interdisciplinary Center for Clinical Research (IZKF) of the medical faculty of Münster (grant Dan3/012/17 to U.D.). This work was further funded in part by the consortia grants from the German Research Foundation (DFG) FOR2107 (DFG grants FOR2107 KI588/14-1, and KI588/14-2, and KI588/20-1, KI588/22-1 to T.K.) and SFB/TRR 393 (project grant no 521379614). S.M. received funding by the (DFG) – Project-ID 521379614 – SFB/TRR 393, as well as ME62262-1, the Else Kröner-Fresenius-Stiftung (grant no 2023_EKEA.153) and the Innovative Medical Research (IMF) of the medical faculty of the University of Münster (grant no ME122205, ME122405). J.G. also received funding from the IMF (GO122301). L.M. received funding by the DFG (493624047 - Clinician Scientist CareerS Münster) and by the IMF (ME122205). T.H. was supported by the German Research Foundation (DFG grants HA7070/2-2, HA7070/3, HA7070/4). The BiDirect study is funded by German Federal Ministry of Education and Research Grants 01ER0816, 01ER1506, and 01ER1205 (K.B.).

## Author contributions

Conceptualization: J.G., N.O., and U.D. Data Acquisition: J.G., N.R.W., S.M., A.K., K.F., L.A., J.K., E.J.L., J.B., L.M.B., T.H., M.G., K.Be., K.Br., F.S., F.T.-O., P.U., L.T., and T.K. Data Preprocessing: D.G., J.B., K.Br., F.S., F.T.-O., P.U., and L.T. Data Curation: J.G., S.M., D.G., A.K., K.F., L.A., E.J.L., M.H., K.Be., K.Br., F.S., F.T.-O., P.U., L.T., and T.K. Resources (providing data/tools/infrastructure): T.H., M.H., K.Be., V.A., B.S., A.J., and T.K. Formal Analysis: J.G., T.H., and U.D. Interpretation of Results: J.G., N.R.W., K.Be., A.J., N.O., and U.D. Writing – Original Draft: J.G. Writing – Review & Editing: N.R.W., S.M., D.G., A.K., K.F., L.A., J.K., E.J.L., J.B., L.M.B., M.R., T.H., L.F., M.G., M.H., K.Be., V.A., K.Br., F.S., F.T-O., P.U., L.T., V.H., H.J., N.A., B.S., A.J., I.N., T.K., N.O., and U.D. Supervision: T.H., V.A., N.O., and U.D. Funding Acquisition: J.G., K.Be., V.A., B.S., T.K., and U.D. All authors reviewed and approved the final version of the manuscript.

## Funding

## Competing interests

T.K. received unrestricted educational grants from Servier, Janssen, Recordati, Aristo, Otsuka, neuraxpharm. M.G. has received remuneration from Janssen for consultancy services. These have no relevance to the work that is covered in the manuscript. The remaining authors declare no competing interests.

## Additional information

[1]Institute for Translational Psychiatry, University of Münster, Münster, Germany. [2]Institute for Translational Neuroscience, University of Münster, Münster, Germany. [3]Clinical Psychology and Translational Psychotherapy, Department of Psychology, University of Münster, Münster, Germany. [4]Clinic for Child and Adolescent Psychiatry, Psychosomatics and Psychotherapy, University Hospital Münster, Münster, Germany. [5]Department of Psychiatry and Neuroscience, Campus Benjamin Franklin, Charité–Universitätsmedizin Berlin, Berlin, Germany. [6]Department of Psychiatry, Psychosomatic Medicine, and Psychotherapy, Goethe University Frankfurt, University Hospital, Frankfurt, Germany. [7]Cooperative Brain Imaging Center - CoBIC, Goethe University Frankfurt, Frankfurt, Germany. [8]Institute of Epidemiology and Social Medicine, University of Münster, Münster, Germany. [9]Department of Psychiatry and Psychotherapy, University of Marburg, Marburg, Germany. [10]Institute of Behavioral Science, Feinstein Institutes for Medical Research, Manhasset, NY, USA. [11]German Center for Mental Health (DZPG), https://www.dzpg.org/en/. [12]Medical School and University Medical Center OWL, Protestant Hospital of the Bethel Foundation, Department of Psychiatry, Bielefeld University, Bielefeld, Germany. [13]These authors contributed equally: Nils Opel, Udo Dannlowski. ✉e-mail: janik.goltermann@gmail.com

