## [Peer Review file · Nature Communications]

Gray matter correlates of childhood maltreatment lack replicability in a multi-cohort brain-wide association study

Corresponding Author: Dr Janik Goltermann

Version 0:

Reviewer comments:

Reviewer #1

(Remarks to the Author)

This is a potentially important study, but I believe that it is fatally flawed in two ways. First, and most important, was the harmonization procedure. They indicated that they used age, sex, total intracranial volume (TIV), and MDD diagnosis as covariates in the harmonization process to preserve variability attributable to these parameters.

“This procedure allows the specification of ‘biological covariates’ that are excluded from harmonization in order to preserve desired variance of potentially confounding variables. We defined age, sex, total intracranial volume (TIV) and MDD diagnosis as such covariates.”

However, it is also critically important to include the outcome of interest as a biological covariate. This is what the user manual states.

“neuroCombat also accepts an optional argument, mod, which is a matrix containing biological covariates, including the outcome of interest. This is recommended to ensure that biological variability is preserved in the harmonization process.”

Hence, by failing to include maltreatment measures as covariates, they likely harmonized out the variability attributable to maltreatment, which can fully explain their null findings.

Second, there are marked sex differences in the associations between maltreatment and brain morphometry (e.g., 1-4). Simply including sex as a covariate will not account for potential differences in the sex-dependent location of abnormalities. Sex-specific analyses are necessary in an effort to assess voxel-base overlap in findings.

The CTQ is also a terribly blunt instrument, although it has been sufficient to identify maltreatment-associated abnormalities in several studies. Much, however, has also been gained from looking at the neuroimaging correlates of exposure to specific types of maltreatment. Further, the CTQ provides no information regarding the timing of exposure, which is also likely a critical determinant (1,5,6), with evidence emerging that prepubertal and postpubertal exposure can have opposite associations with brain function and potentially volume (7,8). This should be discussed in any future version of this study.

1. Teicher, M.H., et al. Differential effects of childhood neglect and abuse during sensitive exposure periods on male and female hippocampus. *Neuroimage* 169, 443-452 (2018).
2. De Bellis, M.D. & Keshavan, M.S. Sex differences in brain maturation in maltreatment-related pediatric posttraumatic stress disorder. *Neurosci Biobehav Rev* 27, 103-117 (2003).
3. Ancelin, M.L., et al. Structural brain alterations in older adults exposed to early-life adversity. *Psychoneuroendocrinology* 129, 105272 (2021).
4. Lugo-Candelas, C., et al. Maternal childhood maltreatment: associations to offspring brain volume and white matter connectivity. *J Dev Orig Health Dis* 14, 591-601 (2023).
5. Ohashi, K., et al. Sex and sensitive period differences in potential effects of maltreatment on axial versus radial diffusivity in the corpus callosum. *Neuropsychopharmacology* 47, 953-964 (2022).

6. Pechtel, P., Lyons-Ruth, K., Anderson, C.M. & Teicher, M.H. Sensitive periods of amygdala development: the role of maltreatment in preadolescence. *Neuroimage* 97, 236-244 (2014).
7. Zhu, J., Anderson, C.M., Ohashi, K., Khan, A. & Teicher, M.H. Potential sensitive period effects of maltreatment on amygdala, hippocampal and cortical response to threat. *Mol Psychiatry* (2023).
8. Zhu, J., et al. Association of Prepubertal and Postpubertal Exposure to Childhood Maltreatment With Adult Amygdala Function. *JAMA Psychiatry* 76, 843-853 (2019).

(Remarks on code availability)

Reviewer #2

(Remarks to the Author)

Review of NCOMMS-24-56581-T

This study uses voxel-based morphometry to examine the association between retrospective reports of childhood maltreatment and grey matter structure in 3 samples of adults with and without diagnoses of major depression diagnoses, with the main findings being 1) a lack of significant associations in the pooled sample of 3225 when controlling for depression diagnosis and 2) a failure to replicate associations observed in one sample in the other 2 samples. A number of different models are tested here. There appears to be a high level of methodological transparency, and the sample sizes are large. However, the theory underlying the analytic decisions and the interpretation of these results is lacking. First, despite it being an increasingly influential model in theories of childhood adversity and neural development, no consideration is taken of the dimensional model of adversity (McLaughlin, Sheridan, et al., multiple citations), which would suggest a different operationalization of adversity than any of the models. The inconclusive results observed in these analyses are largely consistent with what that model would predict. Using aggregates of the “abuse” vs. “neglect” scales could be an additional analysis that may be interesting in this respect. Next, there is almost no acknowledgement of the fact that a retrospective measure of maltreatment in adults is used. Recent work has demonstrated very low agreement between retrospective and prospective measures of maltreatment in adult samples and further that retrospective measures are much more closely linked with psychopathology (Baldwin et al., 2019, 2024). The degree to which and multitude of ways in which retrospective reports of maltreatment are confounded with the depression diagnosis of these participants is therefore a large challenge to the interpretation of all the results that is not sufficiently discussed. Thus, while the challenges of identifying replicable brain-wide associations with other individual differences is an important challenge facing neuroscience, and I would be unsurprised to see the same challenge in the neuroscience of childhood adversity, it is my view that in this case a lack of theory underlying the tested associations in this study undercuts our ability to draw many conclusions from these null findings as currently described and written. More specific comments follow.

Introduction

- P. 3. “In adults, CM effects on gray matter structure have been observed in an array of regions, with most frequent findings implying the hippocampus, amygdala, dorsolateral prefrontal cortex, insula and anterior cingulate cortex (Lim et al., 2014; McLaughlin et al., 2019; Paquola et al., 2016; Pollak et al., 2022; Teicher et al., 2016).” McLaughlin et al. is incorrectly cited as evidence here, as that is a systematic review only of studies of children and adolescents. Moreover, this paper suggests that neither total sum scores on the CTQ, nor dichotomized “maltreatment” are consistently associated with any alterations to brain structure.

Method

- Can the authors please clarify the implications of the image harmonization using the neuroCombat toolbox. What does the defining of age, sex, total intracranial volume (TIV) and MDD diagnosis as “biological covariates” mean in practice, and what influence might this have on results?
- More descriptive statistics on the samples, their age and sex distributions, levels of maltreatment on the various CTQ scales, and severity of depression symptoms would be helpful. Based on the numbers provided in the extreme groups analyses, it seems that depression diagnosis is so confounded with CTQ severity in this sample that the top line findings are not at all surprising.

(Remarks on code availability)

The code appears to be well commented. However, I am not well versed enough in python to effectively review it.

Reviewer #3

(Remarks to the Author)

The manuscript addresses a troubling heterogeneity in findings on the relation between childhood maltreatment (CM) and brain structure alterations. In response to concerns that small sample sizes and publication biases may facilitate the publication of false-positive results, the authors first present a pooled analysis of gray matter volume alterations associated with retrospective reports of CM in adults in a pooled sample of 3,225 individuals from three cohorts. The authors present results from this analysis across a variety of potential modeling strategies (e.g., with and without control for major depressive disorder (MDD), using dichotomous vs continuous measures of CM). The authors then examine these models in each of the three samples and report on the overlap. For the most part, they find that there was little overlap in structural changes associated with CM between samples. This is an excellent topic overdue for serious consideration. The analysis is well

done and the the manuscript well written. There are some limitations to the approach used, especially that the sampling strategies used may emphasize collinearity between CM and MDD, but for the most part they are well characterized. The authors have posted their analysis code according to open-science best practices. Some minor amendments and elaboration on the limitations would improve the manuscript and make it more interpretable and able to stand on its own without reference to the citations:

1. The sole use of retrospective self-reports of childhood maltreatment is an important limitation, and should be discussed in a bit more detail than we see here. The authors present their use of the CTQ as equivalent to other studies in the field. However, several of the papers they cited in support in fact used more robust measurement strategies, notably collecting the CTQ during adolescents when problems with retrospective reporting might be minimized. For example, Gold et. al. used both the CTQ and the Childhood Experiences of Care and Abuse interview and, critically, both were collected during adolescence. Whittle et. al. 2017 also used the CTQ in adolescents, and at two different timepoints. Paquola et. al. also used the measure in adolescents.

2. The cross-sectional nature of the findings are a limitation. Longitudinal studies have been performed on this topic, and should be discussed in the literature review. Specifically, longitudinal research can help clarify confounding effects between MDD and GMV. For example, Weissman et. al. 2020 found that CM sensitized youth through changes in amygdala and hippocampal volume, and that youth who showed this phenotype were more likely to develop depression when exposed to stressful life events. Paquola et. al. also examined these variables in relationship over time. Peverill et al. 2023 found that the association of amygdala volume with CM may be moderated by age.

Peverill, M., Rosen, M. L., Lurie, L. A., Sambrook, K. A., Sheridan, M. A., & McLaughlin, K. A. (2023). Childhood trauma and brain structure in children and adolescents. *Developmental Cognitive Neuroscience*, 59, 101180. <https://doi.org/10.1016/j.dcn.2022.101180>

Weissman, D. G., Lambert, H. K., Rodman, A. M., Peverill, M., Sheridan, M. A., & McLaughlin, K. A. (2020). Reduced hippocampal and amygdala volume as a mechanism underlying stress sensitization to depression following childhood trauma. *Depression and Anxiety*, 37(9), 916–925. <https://doi.org/10.1002/da.23062>

3. As the authors state, the confounding of MDD diagnosis and CM is a central source of ambiguity in these results. This is not a critical weakness -- the authors do present a model without MDD control (model 2), and the lack of replicability shown in this model is striking and interesting. It appears to be the case that each of these samples over-sampled for MDD. It would make sense then, that much of the CM present in the sample would be among the MDD cohorts, and that the two constructs would be highly colinear in this sample. There is little descriptive information about the relationship of CM and MDD presented, but it appears to be the case that only 109 participants were present with severe maltreatment and no MDD diagnosis. The authors seem to imply that the CM results they found could be an artifact of MDD diagnosis (e.g. 367 - 373), but these data are particularly ill suited to tease apart these effects. A few changes would make this more clear:

- a) Please include more descriptive information about the distribution of CTQ scores in the sample, and the relationship of MDD to the CTQ scores.
- b) Please discuss inclusion/exclusion criteria for the samples -- this is important to the interpretation of the results and should not be left to the references.
- c) Could you please include an 'extreme groups' analysis in the pooled sample without control for MDD.

4. The lack of surface based analysis (and metrics such as thickness) is a minor limitation. First, because surface based methods are better able to account for differences in cortical folding between participants, resulting in superior localization (Coalson, et. al. 2018, Tucholka et. al. 2012). Second, because cortical thickness may be especially responsive to early developmental disruption such as CM, and is frequently used in similar studies (e.g., Gold, et. al. (2017), Peverill, et. al. (2023), Ross et. al. (2021). Or see Colich. et. al 2020., for review.) Ultimately, volume and thickness are correlated and I do not expect this to be a critical issue in a sample of this size, but the limitation should be mentioned.

Coalson, T. S., Essen, D. C. V., & Glasser, M. F. (2018). The impact of traditional neuroimaging methods on the spatial localization of cortical areas. *Proceedings of the National Academy of Sciences*, 115(27), E6356–E6365. <https://doi.org/10.1073/pnas.1801582115>

Colich, N. L., Rosen, M. L., Williams, E. S., & McLaughlin, K. A. (2020). Biological aging in childhood and adolescence following experiences of threat and deprivation: A systematic review and meta-analysis. *Psychological Bulletin*, 146(9), 721–764. <https://doi.org/10.1037/bul0000270>

Ross, M.C., Sartin-Tarm, A.S., Letkiewicz, A.M. et al. Distinct cortical thickness correlates of early life trauma exposure and posttraumatic stress disorder are shared among adolescent and adult females with interpersonal violence exposure. *Neuropsychopharmacol.* 46, 741–749 (2021). <https://doi.org/10.1038/s41386-020-00918-y>

Tucholka A, Fritsch V, Poline JB, Thirion B. An empirical comparison of surface-based and volume-based group studies in neuroimaging. *Neuroimage*. 2012 Nov 15;63(3):1443-53. doi: 10.1016/j.neuroimage.2012.06.019. Epub 2012 Jun 22. PMID: 22732555.

5. The sample was limited to 'German Caucasian Individuals'. First, I'm assuming this is a limitation of the data source -- could this please be stated in the manuscript? Second, I understand the term is often used, but the term 'Caucasian' has been criticised because it is derived from outdated, problematic, and inaccurate theories on race. The most likely

explanation is that the sample was restricted in this way because the datasets were limited to participants of European Ancestry to enable valid genetic analysis -- if this is the case then it would be best to say so and 'European Ancestry' would be the preferred way to describe the inclusion criterion.

Shamambo LJ, Henry TL. Rethinking the Use of "Caucasian" in Clinical Language and Curricula: a Trainee's Call to Action. *J Gen Intern Med.* 2022 May;37(7):1780-1782. doi: 10.1007/s11606-022-07431-6. Epub 2022 Feb 25. PMID: 35212875; PMCID: PMC8876079.

Thanks for your work and the opportunity to review,

(Remarks on code availability)

Version 1:

Reviewer comments:

Reviewer #1

(Remarks to the Author)

I appreciate the substantial effort the authors have made to address my previous concerns. However, I still have several critical issues that need to be resolved.

CTQ Scores and Variance Considerations

Including the CTQ sum score is a step in the right direction, but it is insufficient. To fully preserve the variance attributable to each type of maltreatment, the individual CTQ subscale scores should also be included, despite their intercorrelations. The harmonization process should not obscure the unique contributions of different types of maltreatment.

Limitations of Whole-Brain VBM Analysis

The use of whole-brain VBM to detect group differences is problematic, as only highly significant and widespread differences typically survive multiple comparison corrections at the whole-brain level. Given the importance of this negative finding, I recommend complementing the whole-brain approach with region-of-interest (ROI) analyses. Specifically, previously implicated structures such as the hippocampus should be examined using selective masking to provide a more targeted test.

Concerns Regarding Overlapping Voxels

The emphasis on exactly overlapping voxels in VBM analysis is not particularly informative. The VBM pipeline involves morphing individual brains to a standard template, meaning that a given voxel in the template corresponds to slightly different anatomical locations across individuals. A more meaningful approach—consistent with meta-analytic methods—would be to assess effect sizes for maximal differences within specific regions, allowing for a more robust evaluation of effect sizes and study heterogeneity.

Potential Age Confound

Age may be a confounding factor in the reported findings. The effects of maltreatment-related brain alterations may be more discernible in younger adults than in older adults due to neurodevelopmental trajectories. I suggest conducting age-stratified analyses to determine whether these effects are more pronounced within specific age bands.

Medication Status as a Confound

The manuscript does not address the medication status of participants, which could significantly influence the results. Many studies reporting large effects of childhood maltreatment have specifically examined unmedicated samples. It is important to control for medication use in the analysis and, if possible, conduct a separate analysis limited to unmedicated participants.

Limitations of VBM as an Analytic Tool

VBM represents a relatively small proportion of published neuroimaging studies on childhood maltreatment, comprising roughly 15-20% of the literature. Prior analyses, such as those in Teicher et al. (2016), have demonstrated that VBM studies tend to yield less consistent findings compared to other neuroimaging methods. Our research group has discontinued the use of VBM for studying maltreatment-related brain changes due to these limitations. The authors should explicitly acknowledge these limitations in the discussion.

Implications of Total Intracranial Volume (TIV) Correction

The decision to control for TIV warrants further consideration. While TIV correction is generally necessary to account for individual differences in brain size, it assumes that TIV is independent of pathology. In some conditions, such as neurodevelopmental disorders (e.g., autism, ADHD), smaller TIV is an intrinsic feature rather than a confound. In cases where widespread reductions in gray matter volume have been reported (e.g., De Bellis et al.), TIV correction may lead to overcorrection, potentially obscuring true volumetric differences. This issue is particularly relevant when assessing the effects of severe early neglect.

Overall, I recommend that the authors carefully reconsider these issues to strengthen the rigor and interpretability of their findings.

(Remarks on code availability)

Reviewer #2

(Remarks to the Author)

The authors have thoroughly and adequately responded to my comments. I have no further recommendations.

(Remarks on code availability)

Reviewer #3

(Remarks to the Author)

1. The authors have done a much better job describing their cohorts, in particular with respect to levels of CM and depression, and their correlation. However, their writeup still overstates the ability of the analysis to disentangle psychopathology and CM in their sample. Average CTQ scores in the HC group were close to the floor of the instrument, and the rank-biserial r between MDD status and CTQ sum was above .5 in every cohort -- an exceptionally strong confound. If this were a probability sample, this correlation could be considered a relatively natural phenomenon, but the fact that each sample was recruited to over-sample depressed individuals likely inflates this correlation (because the CM we are observing is mostly contingent on someone being in the MDD group), a problem which is further compounded by concerns that the retrospective CM measure could be contaminated by reporting biases associated with symptoms. In their response, the authors argue that this confounding would be more likely to create a false positive than false negative, and I am not sure why. If the model is controlling for a confound correlated with the predictor at .5, it is very likely that the effect of the predictor will be underestimated and/or unstable. They further argue that diagnosis likely drives CM effects because the effects are not visible in their subgroup (within HC, within MDD) analysis. However, they did report effects in their pooled sample in the HC group, despite the fact that their power was greatly limited by the limited amount of CM variability in this group. The absence of an association of CM with gray matter structure in the MDD group is interesting.

In light of the strength of the confounding and the limitations of the study design, my opinion is that the models controlling for MDD diagnosis are of limited interest and should be omitted.

2. Marek et al. (2022) concluded a sample of 9500 would be required to detect a relatively strong brain-wide-association-study effect, 80% of the time, with a corrected p of $p < 10^{-7}$ (which is still more lenient than $.001^3$), and that a sample of 2200 would be required to adequately detect a large effect with an uncorrected $p < .05$. They defined 'replicability' as 'passing the same statistical threshold in sample and out of sample'. In their replication test, the authors' are using an alpha value of .001 (which equates to an effective false negative rate of $.001^3$ or $p \leq 10^{-10}$) with cohort n 's between 557 and 1752. Rather than test each cohort against the others (in sample vs out of sample), they chose to test the overlap between all three cohorts. Could you please better justify the alpha levels defined in the replicability analysis, as well as the choice to compare overlap between three samples rather than out of sample vs in sample for each cohort? As it stands, the lack of replicability you are reporting is easily explained as a power problem in the context of a rather strict alpha threshold. This is still potentially interesting given the even smaller n 's common in the literature you are critiquing (although those are usually paired with more liberal alpha levels), but these choices need justification and discussion.

Minor

- The authors have introduced a new permutation test to test the significance of their replication findings. Could you please provide more details about this test explain why it was added post-hoc?
- In the limitations 'Although depressive states are thought to bias childhood reports,' I believe should be 'depressive states are thought to bias retrospective reports of childhood maltreatment'.

(Remarks on code availability)

Version 2:

Reviewer comments:

Reviewer #3

(Remarks to the Author)

This was an excellent revision. The authors have responded to all of my points constructively and I have no further recommendations.

(Remarks on code availability)

Major revision R1 of the manuscript “Gray matter correlates of childhood maltreatment: searching for replicability in a multi-cohort brain-wide association study” [NCOMMS-24-56581-T] to Nature Communications

We thank the reviewers for their constructive feedback. Further, we thank the editor for the possibility to revise and resubmit our manuscript. We have now made substantial changes and extensions according to the reviewers’ suggestions. Specifically, we reran the Combat harmonization procedure and all analyses to make sure that the initial omission of CM as a ‘biological covariate’ did not bias our results. Furthermore, we added sex-stratified analyses, as well as maltreatment subtype analyses differentiating abuse and neglect subscales as justified by a prominent theoretical model (the dimensional model of adversity). In addition, we obtained permutation-based null distributions for voxel overlap in effects across cohorts (how much overlap is observed by chance) to get formalized significance estimates for our replicability results. In summary, the additional analyses lead to a highly similar pattern of non-replicability of effects as the initial analyses, thus not changing the overall narrative of the manuscript. Overall, we present a total of 180 brain-wide statistical analyses that have consistently shown non-replicability of gray matter correlates of CM (15 statistical models, across pooled cohorts and all three cohorts separately, and finally rerun stratified by sex). As several reviewers have stressed the limitations of the CTQ as an assessment instrument, we have extended the CTQ-related limitation discussion and have prominently stated within the abstract that maltreatment was measured retrospectively by the CTQ to enhance interpretability of our findings. Finally, we reformatted the manuscript according to the journal formatting guidelines. We believe that we were able to fully address the major concerns of the reviewers and that the additions have substantially improved the manuscript, further strengthening its conclusions. A reference list specifically for the rebuttal is provided at the end of this document. We provide revised documents with changes highlighted.

REVIEWER COMMENTS

Reviewer #1 (Remarks to the Author):

This is a potentially important study, but I believe that it is fatally flawed in two ways. First, and most important, was the harmonization procedure. They indicated that they used age, sex, total intracranial volume (TIV), and MDD diagnosis as covariates in the harmonization process to preserve variability attributable to these parameters.

“This procedure allows the specification of ‘biological covariates’ that are excluded from harmonization in order to preserve desired variance of potentially confounding variables. We defined age, sex, total intracranial volume (TIV) and MDD diagnosis as such covariates.”

However, it is also critically important to include the outcome of interest as a biological covariate. This is what the user manual states.

“neuroCombat also accepts an optional argument, mod, which is a matrix containing biological covariates, including the outcome of interest. This is recommended to ensure that biological variability is preserved in the harmonization process.”

Hence, by failing to include maltreatment measures as covariates, they likely harmonized out the variability attributable to maltreatment, which can fully explain their null findings.

This is a very important point, and we fully acknowledge this error in the harmonization procedure. In order to make sure that this omission did not drive our non-replicability in results we reran the harmonization including CTQ sum as an additional biological covariate. This inclusion did not qualitatively alter any results. While, in principle, the omission of CTQ as a biological covariate could lead to CM-related gray matter variability to be “harmonized out”, there are several reasons why this does not seem to apply to our case: this problem would only happen 1) if CTQ variance is substantially confounded with scanner hardware groups and 2) if this confounding is not sufficiently accounted for in the biological covariates already included in the harmonization procedure. There were detectable differences in CM severity across cohorts (see Table S2-S3). We ran an ANOVA to test these differences specifically across scanner groups (the six groups also included in the harmonization procedure) and found that the scanner groups explain only 0.3% of variance in CTQ sum scores ($\eta^2=.003$). While this is significant in this large sample ($p=.031$), we believe that the reason why the additional inclusion of CTQ sum as a biological covariate did not substantially alter the results is that these differences in CM severity between scanner groups are negligible small. Furthermore, it is possible that the prior inclusion of the other biological covariates already accounted for some of this remaining variance due to the association between CTQ sum and these variables (particularly with MDD diagnosis). Nevertheless, we agree that including CTQ sum as a covariate in the Combat harmonization process is the correct way of applying the method and thus have changed the methods accordingly on p. 16 of the main text, and consequently, have updated all obtained results. Furthermore, we added a more detailed explanation of the rationale of including biological covariates in the harmonization in the supplements (p. 5) due to a comment by Reviewer #2 (see below).

Second, there are marked sex differences in the associations between maltreatment and brain morphometry (e.g., 1-4). Simply including sex as a covariate will not account for potential differences in the sex-dependent location of abnormalities. Sex-specific analyses are necessary in an effort to assess voxel-base overlap in findings.

Thank you for this valuable remark. It is certainly true that sex-specific effects have received a lot of attention within previous studies, and we agree that the evidence warrants a more detailed investigation of this matter (particularly, due to our null findings). We agree that solely adding sex as a covariate is not sufficient to account for this. Controlling for sex only protects from maltreatment effects being driven by confounding sex effects but does not test for maltreatment effects which may be specific for male or female participants. Thus, we decided to follow your suggestion and conducted robustness analyses, rerunning all models stratified by sex. Consequently, we added a short introduction for previous findings on sex-specific neural correlates of CM on p. 3, including one of the references suggested by you (Teicher et al., 2018), as well as two other studies reporting on similar sex-specific effects (Everaerd et al., 2016; Frodl et al., 2017). In summary, rerunning all analyses separately for female and male subsamples did not substantially alter our results. While we observed descriptively somewhat larger overlap in female subsamples, the overall pattern of non-replicability was consistent. As indicated by our newly installed permutation-based significance test, the observed overlap in all analyses was not higher than expected by chance. This was consistent across all statistical models and across all analyses stratified by sex. We believe that these additional analyses further strengthen the overall conclusion of non-replicability of childhood maltreatment associations with gray matter alterations, under the premise of the utilized methods. All sex stratified results for pooled analyses are provided in supplementary Tables S6-S7. Cohort-wise results stratified by sex are shown in Tables S9-S10 and S12-S13, as well as in Figures S21-S50. Replicability

results (i.e., overlap in results across cohorts) stratified by sex are shown in Tables S14-S15 and S17-S18. The sex-specific results are briefly summarized within the main text on pp. 5 and 10 in the results section and on pp. 11 and 12 in the discussion section.

The CTQ is also a terribly blunt instrument, although it has been sufficient to identify maltreatment-associated abnormalities in several studies. Much, however, has also been gained from looking at the neuroimaging correlates of exposure to specific types of maltreatment. Further, the CTQ provides no information regarding the timing of exposure, which is also likely a critical determinant (1,5,6), with evidence emerging that prepubertal and postpubertal exposure can have opposite associations with brain function and potentially volume (7,8). This should be discussed in any future version of this study.

We agree with the reviewer on the limitations of the CTQ as an assessment instrument to measure childhood maltreatment. It has been criticized for a number of reasons, including the mentioned omission of the timing of exposure which has emerged as a promising potential moderator for neurobiological and clinical consequences of CM. While we mentioned this limitation of the CTQ briefly within the discussion section already in the previous version of the manuscript, we have now expanded on this important limitation, citing several studies reporting timing effects for the association between CM and clinical, as well as neurobiological endpoints (including the studies suggested by the reviewer). While the availability of more detailed assessments of CM is of course desirable, there is an inherent tradeoff between assessment depth and sample size. The CTQ has the advantage of being efficient to apply. This enabled us to accumulate these unprecedented large sample sizes with this measure being available. Again, we would like to point out that despite all the criticism of the CTQ, previous evidence about neural correlates of CM is based largely on studies measuring these constructs via the CTQ (with exceptions as pointed out in the revised discussion section on p. 12). Thus, our methods are highly comparable with the majority of studies investigating this matter and the discrepancy in findings (non-replicability in our large samples) seems relevant. This even includes findings presented by our own group based on earlier stages of the MNC dataset, where CTQ scales were associated with brain structure. But this finding now turns out to not be robust in larger samples. We agree that it would be good if similar findings based on other assessment instruments (such as the MACE interview) were subject for replication efforts as well. Unfortunately, replications are very rare in the whole field of psychiatric neuroimaging. With our analysis regarding the CTQ we simply aim to take a first step towards replicating central findings of the field.

Regarding subtypes of maltreatment, the CTQ provides admittedly quite short subscales to capture information on emotional, physical and sexual abuse, as well as physical and emotional neglect. Although these subscales only consist of five items each, they have repeatedly shown to have relatively good psychometric qualities (Bernstein et al., 2003; Hernandez et al., 2013; Wingenfeld et al., 2010). Upon your comment and a remark by Reviewer #2 we have now expanded the introduction regarding the theoretical framework and rationale regarding the investigation of subtype-specific neural correlates. We have incorporated a short introduction into the dimensional model of adversity (McLaughlin et al., 2019) on p. 3, and have accordingly added further analyses testing specific effects of threat-related CM subtypes (the CTQ abuse subscales) and deprivation-related CM subtypes (the CTQ neglect subscales). These additional analyses provided a pattern of results consistent with previous models. Again, we agree that other assessment instruments are available that provide more fine-grained assessments of CM

types (such as the mentioned MACE interview). This remains a limitation of the current work which we believe is now becoming clearer within the discussion section (p. 12).

1. Teicher, M.H., et al. Differential effects of childhood neglect and abuse during sensitive exposure periods on male and female hippocampus. *Neuroimage* 169, 443-452 (2018).
2. De Bellis, M.D. & Keshavan, M.S. Sex differences in brain maturation in maltreatment-related pediatric posttraumatic stress disorder. *Neurosci Biobehav Rev* 27, 103-117 (2003).
3. Ancelin, M.L., et al. Structural brain alterations in older adults exposed to early-life adversity. *Psychoneuroendocrinology* 129, 105272 (2021).
4. Lugo-Candelas, C., et al. Maternal childhood maltreatment: associations to offspring brain volume and white matter connectivity. *J Dev Orig Health Dis* 14, 591-601 (2023).
5. Ohashi, K., et al. Sex and sensitive period differences in potential effects of maltreatment on axial versus radial diffusivity in the corpus callosum. *Neuropsychopharmacology* 47, 953-964 (2022).
6. Pechtel, P., Lyons-Ruth, K., Anderson, C.M. & Teicher, M.H. Sensitive periods of amygdala development: the role of maltreatment in preadolescence. *Neuroimage* 97, 236-244 (2014).
7. Zhu, J., Anderson, C.M., Ohashi, K., Khan, A. & Teicher, M.H. Potential sensitive period effects of maltreatment on amygdala, hippocampal and cortical response to threat. *Mol Psychiatry* (2023).
8. Zhu, J., et al. Association of Prepubertal and Postpubertal Exposure to Childhood Maltreatment With Adult Amygdala Function. *JAMA Psychiatry* 76, 843-853 (2019).

Reviewer #2 (Remarks to the Author):

Review of NCOMMS-24-56581-T

This study uses voxel-based morphometry to examine the association between retrospective reports of childhood maltreatment and grey matter structure in 3 samples of adults with and without diagnoses of major depression diagnoses, with the main findings being 1) a lack of significant associations in the pooled sample of 3225 when controlling for depression diagnosis and 2) a failure to replicate associations observed in one sample in the other 2 samples. A number of different models are tested here. There appears to be a high level of methodological transparency, and the sample sizes are large. However, the theory underlying the analytic decisions and the interpretation of these results is lacking. First, despite it being an increasingly influential model in theories of childhood adversity and neural development, no consideration is taken of the dimensional model of adversity (McLaughlin, Sheridan, et al., multiple citations), which would suggest a different operationalization of adversity than any of the models. The inconclusive results observed in these analyses are largely consistent with what that model would predict. Using aggregates of the “abuse” vs. “neglect” scales could be an additional analysis that may be interesting in this respect.

This is an important remark. Previous studies on neural correlates of CM have used a number of different operationalizations of CM and its subtypes, and these distinctions often lack valid theoretical rationales. The dimensional model of adversity is a well-studied theoretical framework and we thank the reviewer for this intriguing suggestion. While any differential effects of abuse and neglect could already have shown within the analysis of the five subscales that we conducted, we agree that it is possible that an aggregation of the three abuse subscales and the two neglect subscales could add psychometric quality to these constructs, when assuming the dimensional model of adversity by McLaughlin, Sheridan et al. Thus, we have now added a short introduction of this framework and associated references (p. 3) and based on

this rationale have conducted additional analyses testing aggregate “abuse” and “neglect” scores in additional models (in the current revised version these are models 5 and 6). We have updated the methods part (p. 17) and Table 1 accordingly. These analyses yielded null results comparable to previous approaches investigating the CTQ subscales. We decided to still present the results of the five CTQ subscales in the paper, in addition to the ‘abuse’/‘neglect’ analyses for the following reason: some scholars have suggested that a more detailed differentiation of CM subtypes could be important and could yield specific neural correlates, such as for example sexual abuse affecting the development of somatosensory cortices or verbal abuse influencing the auditory cortex (Teicher et al., 2016). Thus, we believe that testing this more fine-grained differentiation of CM subtypes is warranted due to previous studies and adds value to the presented null-findings. We want to emphasize that we strongly agree that there is substantial theoretical foundation for the postulation of subtype-specific neural correlates but at the same time the practical investigation is challenging due to the empirical co-occurrence of different types of CM (as can be seen for our specific samples in our new Figure 1a and Figures S2-S4). This limits the conclusions we are able to draw from our subtype-specific analyses.

Next, there is almost no acknowledgement of the fact that a retrospective measure of maltreatment in adults is used. Recent work has demonstrated very low agreement between retrospective and prospective measures of maltreatment in adult samples and further that retrospective measures are much more closely linked with psychopathology (Baldwin et al., 2019, 2024). The degree to which and multitude of ways in which retrospective reports of maltreatment are confounded with the depression diagnosis of these participants is therefore a large challenge to the interpretation of all the results that is not sufficiently discussed. Thus, while the challenges of identifying replicable brain-wide associations with other individual differences is an important challenge facing neuroscience, and I would be unsurprised to see the same challenge in the neuroscience of childhood adversity, it is my view that in this case a lack of theory underlying the tested associations in this study undercuts our ability to draw many conclusions from these null findings as currently described and written. More specific comments follow.

We thank the reviewer for bringing up this important point. The work by Baldwin et al. is indeed very interesting and raises questions about what the CTQ actually measures (along with other retrospective instruments of CM). In our previous version of the manuscript, we have only briefly mentioned this. We have now revised this part of the discussion section (p. 12) and expanded on the potential limitations of the CTQ. We have also added a reference to our own study investigating a potential depressive bias on CTQ reports (Goltermann et al., 2023). In this study we showed that CM reports using the CTQ are highly stable across a 2-year interval, in non-clinical as well as MDD samples. Importantly, the MDD sample did not show systematic associations between changes in depression severity and changes in retrospective CM reports (i.e., individuals becoming more depressed over time did not systematically score higher in the CTQ). Thus, we would argue that an undue bias of the CTQ due to depression seems to be negligible. However, limitations regarding the utilization of such retrospective self-report instruments remain, and we discuss these limitations in more detail in our revised discussion section (see p. 12).

We also agree that the confounding between CM and psychopathology poses a substantial challenge when investigating neural correlates of either construct. Previous studies have handled this problem very differently, for example by using psychiatric diagnosis as a control variable or by investigating subgroups. We feel that each approach by itself comes with important limitations: Controlling for diagnosis presumably ‘takes away’ substantial variance of the predictor, thus making it more difficult to identify effects. On the other hand, not controlling for diagnosis makes a differentiation of diagnosis effects vs. CM effects

impossible. Subgroup analyses of non-clinical and clinical samples have the disadvantage of constraining variance of the predictor but allow better disentanglement of diagnosis versus CM effects. Notably, even in subgroup analyses of MDD samples clinical variables, such as current depression severity are still confounded with CM and could drive effects. While positive findings (identifying significant, replicable associations) could be explainable by this confounding, it seems unlikely that the residual confounding between CM and clinical characteristics could have somehow driven our extensive null findings. These were surprisingly consistent across all analyses, with our statistical models mirroring the methodologies of previous studies reporting on positive findings. We used methods highly comparable to those by previous studies, reporting on CM-GMV associations (e.g., as Dannlowski et al., 2012 using HC subgroup analysis). For further context we have expanded our results by systematic investigations of the relationship between CM and clinical variables in our data (Figure 1a-b and Figure S2-S5). As expected, we see quite clear associations between CTQ scales and clinical variables which are replicable across all cohorts. We have added a summary of these associations within a distinct subsection within the results of the main text (p. 5). Finally, we acknowledge that the utilization of the CTQ implies important limitation due to the retrospective and subjective nature of its assessments. This remains to be a limitation of our study, and we believe that our expanded discussion now more adequately accounts for this matter. Furthermore, we now prominently include the information that CM was retrospectively measured using the CTQ within the abstract, and have pointed this out as a central limitation within the first paragraph of the discussion section (p. 11) to increase the interpretability of our findings.

Introduction

- P. 3. “In adults, CM effects on gray matter structure have been observed in an array of regions, with most frequent findings implying the hippocampus, amygdala, dorsolateral prefrontal cortex, insula and anterior cingulate cortex (Lim et al., 2014; McLaughlin et al., 2019; Paquola et al., 2016; Pollok et al., 2022; Teicher et al., 2016).” McLaughlin et al. is incorrectly cited as evidence here, as that is a systematic review only of studies of children and adolescents. Moreover, this paper suggests that neither total sum scores on the CTQ, nor dichotomized “maltreatment” are consistently associated with any alterations to brain structure.

We apologize for this oversight and thank the reviewer for the important comment. We have removed the respective reference here. The reference is still cited within the main text when introducing the dimensional model of adversity, regarding the evidence for neural correlates of threat and deprivation experiences (now noting that this review summarized evidence of studies including children and adolescents; p. 3).

Method

- Can the authors please clarify the implications of the image harmonization using the neuroCombat toolbox. What does the defining of age, sex, total intracranial volume (TIV) and MDD diagnosis as “biological covariates” mean in practice, and what influence might this have on results?

The inclusion of biological covariates in Combat is recommended in the toolbox manual to avoid that variability within gray matter intensities that is of interest (i.e., associated with maltreatment or other covariates included in the utilized models) is harmonized out in the harmonization process. Thus, variables for which associated variability with the biological outcome should be preserved can be specified, while variability due to scanner differences are neutralized. A simple example: if I have an old sample X measured at scanner x and a young sample Y measured at scanner y then the Combat algorithm will harmonize (i.e., make even)

differences in brain variables between the cohorts and thus will also neutralize any variance that is shared between scanner groups and age. To avoid this problem, 'biological covariates' (in this example age) are defined that are excluded from the harmonization procedure. The influence this has on the results depends on 1) the degree to which scanner groups are confounded with the respective biological covariate and 2) the strength of the association between the biological covariate and the biological outcome (in this case the gray matter intensities). For instance, the inclusion of age as a covariate is particularly important because age is highly confounded with cohort in our analysis. Participants from the BiDirect cohort are substantially older (mean age 51.7 years) as compared to the other two cohorts (MACS: 35.3 years; MNC: 35.4 years). In addition, age effects on gray matter are particularly large and importantly much larger than expected effect sizes associated with psychiatric variables (Good et al., 2001; Marek et al., 2022; Winter et al., 2022). If neglected in the harmonization procedure, the large age effects on gray matter would have been partly attributed to confounding scanner differences and associated gray matter variability would have been eliminated, thus biasing results. On the other hand, the inclusion of CTQ sum in addition to the other covariates did not qualitatively alter our results as described above in the reply to Reviewer #1. This was presumably due to only very small differences in CTQ sum across scanner groups and in addition, because portions of this variability were already accounted for by including MDD diagnosis and the other covariates. However, as Reviewer #1 correctly pointed out the inclusion of the main predictor of interest is standardly recommended. Thus, we added CTQ sum as an additional biological covariate in the harmonization procedure and reran all analyses and updated all results. We have now added a more detailed description and rationale of this harmonization procedure within the supplements (p. 5).

- More descriptive statistics on the samples, their age and sex distributions, levels of maltreatment on the various CTQ scales, and severity of depression symptoms would be helpful. Based on the numbers provided in the extreme groups analyses, it seems that depression diagnosis is so confounded with CTQ severity in this sample that the top line findings are not at all surprising.

Detailed sample characteristics, including age and sex distributions, levels of maltreatment and several clinical characteristics of the MDD subsamples (including depressive symptom severity) are given stratified by cohorts and by diagnostic groups within Table S1 and Table S2. Cohort differences in these characteristics were formally tested and are reported in Table S3 and Table S4. We have now slightly revised the methods section to point out more explicitly what information is provided in these tables (p. 14). Upon your comment we have now also added a new supplementary Figure S1 showing the age distribution across cohorts and diagnosis groups. Based on your comment, as well as a comment by Reviewer #3 we have taken the chance to expand our results regarding the relationship between CM and depression in our data. We tested to what extent CTQ scales, demographic variables and clinical variables are associated with each other, using Spearman correlations and Mann-Whitney-U tests (due to the non-normal distributions of the CTQ scales). Here we found that participants with MDD reported significantly higher CM severity as compared to HC samples across all CTQ subscales and consistently in all cohorts (largest effects found for the emotional neglect subscale: $r=.517$). Furthermore, within the MDD samples there was a general pattern of CTQ scales being positively associated with clinical severity (higher current symptom severity, earlier age of onset and higher numbers of previous depressive episodes and psychiatric inpatient treatments). This was mainly consistent across cohorts. We have added a short description of these analyses within the methods section under 'Statistical analysis' (p. 16), added a short

paragraph reporting these results within the results section (p. 5), and have added a short discussion of these results contrasting the replicability of the link between CM and psychopathology with the non-replicability of the brain-related effects of CM (p. 11).

Reviewer #2 (Remarks on code availability):

The code appears to be well commented. However, I am not well versed enough in python to effectively review it.

Reviewer #3 (Remarks to the Author):

The manuscript addresses a troubling heterogeneity in findings on the relation between childhood maltreatment (CM) and brain structure alterations. In response to concerns that small sample sizes and publication biases may facilitate the publication of false-positive results, the authors first present a pooled analysis of gray matter volume alterations associated with retrospective reports of CM in adults in a pooled sample of 3,225 individuals from three cohorts. The authors present results from this analysis across a variety of potential modeling strategies (e.g., with and without control for major depressive disorder (MDD), using dichotomous vs continuous measures of CM). The authors then examine these models in each of the three samples and report on the overlap. For the most part, they find that there was little overlap in structural changes associated with CM between samples. This is an excellent topic overdue for serious consideration. The analysis is well done and the manuscript well written. There are some limitations to the approach used, especially that the sampling strategies used may emphasize collinearity between CM and MDD, but for the most part they are well characterized. The authors have posted their analysis code according to open-science best practices. Some minor amendments and elaboration on the limitations would improve the manuscript and make it more interpretable and able to stand on its own without reference to the citations:

1. The sole use of retrospective self-reports of childhood maltreatment is an important limitation, and should be discussed in a bit more detail than we see here. The authors present their use of the CTQ as equivalent to other studies in the field. However, several of the papers they cited in support in fact used more robust measurement strategies, notably collecting the CTQ during adolescents when problems with retrospective reporting might be minimized. For example, Gold et. al. used both the CTQ and the Childhood Experiences of Care and Abuse interview and, critically, both were collected during adolescence. Whittle et. al. 2017 also used the CTQ in adolescents, and at two different timepoints. Paquola et. al. also used the measure in adolescents.

We agree that the utilization of the CTQ as an assessment instrument comes with important limitations to the findings. As described above, as a response to the other reviewers, we have now expanded our discussion on the limitations of the CTQ (p. 12), including the neglect of potentially important information on the timing of exposure of CM experiences, as well as the CTQ as a retrospective self-report questionnaire, which may lead to different results than using prospective measures. While the majority of studies have indeed used the CTQ, we appreciate that the reviewer has pointed us towards the fact that numerous of the available findings from previous studies are based on other assessment instruments. Thus, we have added the references suggested by the reviewer and are now putting more emphasis within the discussion section on the limitation that the utilization of CTQ has on the generalizability of the findings (p. 12). As mentioned above as a reply to Reviewer #1 we do not claim that our results

necessarily refute findings on neural correlates of CM that are based on other assessment instruments. We merely attempt to make a first step in investigating replicability of brain findings in the field of mental health. Future studies that expand this replication framework to other operationalizations/measurements of CM and also to other related research questions would be desirable and are in our opinion necessary. Thus, we formulated this conclusion within the discussion section on p. 12. Furthermore, we now prominently include the information that CM was retrospectively measured using the CTQ within the abstract, and have pointed this out as a central limitation within the first paragraph of the discussion section (p. 11) to increase the interpretability of our findings.

2. The cross-sectional nature of the findings are a limitation. Longitudinal studies have been performed on this topic, and should be discussed in the literature review. Specifically, longitudinal research can help clarify confounding effects between MDD and GMV. For example, Weissman et. al. 2020 found that CM sensitized youth through changes in amygdala and hippocampal volume, and that youth who showed this phenotype were more likely to develop depression when exposed to stressful life events. Paquola et. al. also examined these variables in relationship over time. Peverill et al. 2023 found that the association of amygdala volume with CM may be moderated by age.

Thank you for this important remark. Of course, our findings are limited to the cross-sectional design utilized in our study. It is possible that neural correlates of CM are more replicable in a longitudinal design. While longitudinal studies are quite sparse in this domain the reviewer mentions important studies that have used such a longitudinal design. We have added a short section to discuss this limitation on p. 12 and have added some of the suggested references. Again, it remains to be investigated in future studies if the replicability is higher for this domain/design. In our opinion it would be highly desirable to expand our replication framework to other methodologies on the same topic (and to other research questions in psychiatric neuroimaging in general). We make this point clearer now within the discussion section.

Peverill, M., Rosen, M. L., Lurie, L. A., Sambrook, K. A., Sheridan, M. A., & McLaughlin, K. A. (2023). Childhood trauma and brain structure in children and adolescents. *Developmental Cognitive Neuroscience*, 59, 101180. <https://doi.org/10.1016/j.dcn.2022.101180>

Weissman, D. G., Lambert, H. K., Rodman, A. M., Peverill, M., Sheridan, M. A., & McLaughlin, K. A. (2020). Reduced hippocampal and amygdala volume as a mechanism underlying stress sensitization to depression following childhood trauma. *Depression and Anxiety*, 37(9), 916–925. <https://doi.org/10.1002/da.23062>

3. As the authors state, the confounding of MDD diagnosis and CM is a central source of ambiguity in these results. This is not a critical weakness -- the authors do present a model without MDD control (model 2), and the lack of replicability shown in this model is striking and interesting. It appears to be the case that each of these samples over-sampled for MDD. It would make sense then, that much of the CM present in the sample would be among the MDD cohorts, and that the two constructs would be highly colinear in this sample. There is little descriptive information about the relationship of CM and MDD presented, but it appears to be the case that only 109 participants were present with severe maltreatment and no MDD diagnosis. The authors seem to imply that the CM results they found could be an artifact of MDD diagnosis (e.g. 367 - 373), but these data are particularly ill suited to tease apart these effects.

This is indeed a very important point and represents one of the major challenges when investigating the neural correlates of CM and their association with psychopathology. Thus, it

is not surprising that multiple reviewers have commented on this (also see reply to reviewer #2). The notion, that underlying diagnosis effects drive our apparent CM effects in models not controlling for MDD comes from several observations: 1) the CM effects are strongest when diagnosis is not controlled for, 2) if these effects would be driven by CM, similar effects should become apparent in HC subsamples or at least in MDD subsamples, 3) when focusing on lifetime MDD as the primary predictor, as done in another recent analysis by our group using the same cohorts, similar clusters emerge as in our current Model 2 and Model 13 but their associations with MDD diagnosis are highly replicable across cohorts, partly even at FWE-corrected significance levels (Dannlowski et al., 2024). This is shortly addressed within the discussion section on p. 12. In addition, we have decided to follow the reviewer's suggestions to add considerably more information on the nature of the confounding/association between CM and psychopathology in our data (see replies below).

A few changes would make this more clear:

a) Please include more descriptive information about the distribution of CTQ scores in the sample, and the relationship of MDD to the CTQ scores.

Some information about this was already given in the previous version of our manuscript. There, we presented descriptive statistics for all CTQ scales across all cohorts, separately for HC and MDD groups (Table S2). We have now expanded this by testing the difference in CTQ scales across these groups statistically in Table S5. These analyses yielded a clear pattern of higher CM reports in the MDD group, as compared to the HC group. This was consistent across all subscales and all cohorts (maximum effect size in pooled sample: $r=.517$ for the emotional neglect subscale). Furthermore, we have added a heatmap-style correlation matrix to give an overview about the associations between CTQ scales and demographic and (in MDD samples) clinical variables. This is shortly described within the analysis section (p. 16) and results are shown in the current Figure 1a. To provide further information on the distribution of CTQ scores across HC and MDD groups we also added violin plots in Figure 1b. Both subplots from Figure 1 are furthermore given for each cohort separately in supplementary Figures S2-S5.

b) Please discuss inclusion/exclusion criteria for the samples -- this is important to the interpretation of the results and should not be left to the references.

Upon your request we have now moved the exclusion/inclusion criteria from the supplements to the main text (p. 14). We agree that this adjustment makes this important information much more explicit and available for readers, thus improving the interpretability of findings. We have also added the low ethnic variability of our samples as an important limitation for all conclusions. This was added prominently within the first paragraph of the discussion section (p. 11) and is also discussed in an additional short section on p. 13.

c) Could you please include an 'extreme groups' analysis in the pooled sample without control for MDD.

We have followed this suggestion and have added a model comparing severe CM with 'none to minimal' CM in a sample including HC and MDD individuals, while not controlling for diagnosis. In order to stay consistent in our analysis plan we fitted this model in the pooled sample, as well as in each cohort separately, enabling us to investigate the replicability of its effects also. As can be expected, this model yielded widespread clusters in each cohort (and in the pooled cohorts), similar to Model 2 (CTQ sum in HC/MDD without controlling for diagnosis). However, our newly introduced permutation tests indicated that even these relatively large effects were not significantly replicable.

4. The lack of surface based analysis (and metrics such as thickness) is a minor limitation. First, because surface based methods are better able to account for differences in cortical folding between participants, resulting in superior localization (Coalson, et. al. 2018, Tucholka et. al. 2012). Second, because cortical thickness may be especially responsive to early developmental disruption such as CM, and is frequently used in similar studies (e.g., Gold, et. al. (2017), Peverill, et. al. (2023), Ross et. al. (2021). Or see Colich. et. al 2020., for review.) Ultimately, volume and thickness are correlated and I do not expect this to be a critical issue in a sample of this size, but the limitation should be mentioned.

Thank you for this important remark. We agree that the replicability could potentially differ in other imaging modalities. This also includes for example fMRI- and DTI-based neural correlates of CM but of course this does also imply other metrics of gray matter structure, such as measures of cortical surface or thickness. We have added this limitation in a paragraph within the discussion section, together with discussing generalizability to other MRI modalities (p. 13).

Coalson, T. S., Essen, D. C. V., & Glasser, M. F. (2018). The impact of traditional neuroimaging methods on the spatial localization of cortical areas. *Proceedings of the National Academy of Sciences*, 115(27), E6356–E6365. <https://doi.org/10.1073/pnas.1801582115>

Colich, N. L., Rosen, M. L., Williams, E. S., & McLaughlin, K. A. (2020). Biological aging in childhood and adolescence following experiences of threat and deprivation: A systematic review and meta-analysis. *Psychological Bulletin*, 146(9), 721–764. <https://doi.org/10.1037/bul0000270>

Ross, M.C., Sartin-Tarm, A.S., Letkiewicz, A.M. et al. Distinct cortical thickness correlates of early life trauma exposure and posttraumatic stress disorder are shared among adolescent and adult females with interpersonal violence exposure. *Neuropsychopharmacol.* 46, 741–749 (2021). <https://doi.org/10.1038/s41386-020-00918-y>

Tucholka A, Fritsch V, Poline JB, Thirion B. An empirical comparison of surface-based and volume-based group studies in neuroimaging. *Neuroimage*. 2012 Nov 15;63(3):1443-53. doi: 10.1016/j.neuroimage.2012.06.019. Epub 2012 Jun 22. PMID: 22732555.

5. The sample was limited to 'German Caucasian Individuals'. First, I'm assuming this is a limitation of the data source -- could this please be stated in the manuscript? Second, I understand the term is often used, but the term 'Caucasian' has been criticised because it is derived from outdated, problematic, and inaccurate theories on race. The most likely explanation is that the sample was restricted in this way because the datasets were limited to participants of European Ancestry to enable valid genetic analysis -- if this is the case then it would be best to say so and 'European Ancestry' would be the preferred way to describe the inclusion criterion.

We thank the reviewer for this important comment. The inclusion of 'German Caucasian individuals' is indeed already a study policy at the step of participant recruitment (particularly to obtain genetically homogeneous samples for genetic analyses). Thus, we have moved this information from the analysis-specific exclusion criteria to the more general description of the cohorts (p. 14), there we have also added a short rationale for obtaining genetically homogenous samples. Furthermore, we agree that the term 'Caucasian' is indeed outdated and problematic. We have now changed this to "western European ancestry" throughout the manuscript as this describes the recruitment policy most accurately.

Shamambo LJ, Henry TL. Rethinking the Use of "Caucasian" in Clinical Language and Curricula: a Trainee's Call to Action. *J Gen Intern Med*. 2022 May;37(7):1780-1782. doi: 10.1007/s11606-022-

07431-6. Epub 2022 Feb 25. PMID: 35212875; PMCID: PMC8876079.

Thanks for your work and the opportunity to review,
Matthew Peverill, Ph.D.

References by the manuscript's authors:

- Bernstein, D. P., Stein, J. A., Newcomb, M. D., Walker, E., Pogge, D., Ahluvalia, T., Stokes, J., Handelsman, L., Medrano, M., Desmond, D., & Zule, W. (2003). Development and validation of a brief screening version of the Childhood Trauma Questionnaire. *Child Abuse and Neglect*, 27(2), 169–190. [https://doi.org/10.1016/S0145-2134\(02\)00541-0](https://doi.org/10.1016/S0145-2134(02)00541-0)
- Dannowski, U., Stuhrmann, A., Beutelmann, V., Zwanzger, P., Lenzen, T., Grotegerd, D., Domschke, K., Hohoff, C., Ohrmann, P., Bauer, J., Lindner, C., Postert, C., Konrad, C., Arolt, V., Heindel, W., Suslow, T., & Kugel, H. (2012). Limbic Scars: Long-Term Consequences of Childhood Revealed by Functional and Structural Magnetic Resonance Imaging. *Biological Psychiatry*, 71(4), 286–293. <https://doi.org/10.1016/j.biopsych.2011.10.021>
- Dannowski, U., Winter, N. R., Meinert, S., Grotegerd, D., Kraus, A., Flinkenflügel, K., Leehr, E., Böhnlein, J., Borgers, T., Fisch, L., Bauer, M., Pfennig, A., Richter, M., Opel, N., Repple, J., Gruber, M., Minnerup, H., Hermesdorf, M., Nitsch, R., ... Goltermann, J. (2024). Replicability and generalizability of gray matter reductions in major depression: a voxel-based investigation of 4021 individuals. *SSRN (Preprint)*. <https://doi.org/http://dx.doi.org/10.2139/ssrn.4854882>
- Everaerd, D., Klumpers, F., Zwiers, M., Guadalupe, T., Franke, B., Van Oostrom, I., Schene, A., Fernández, G., & Tendolkar, I. (2016). Childhood abuse and deprivation are associated with distinct sex-dependent differences in brain morphology. *Neuropsychopharmacology*, 41(7), 1716–1723. <https://doi.org/10.1038/npp.2015.344>
- Frodal, T., Janowitz, D., Schmaal, L., Tozzi, L., Dobrowolny, H., Stein, D. J., Veltman, D. J., Wittfeld, K., Erp, T. G. M. Van, Jahanshad, N., Block, A., Hegenscheid, K., Lagopoulos, J., Hatton, S. N., Hickie, I. B., Maria, E., Carballido, A., Brooks, S. J., Vuletic, D., ... Grabe, H. J. (2017). Childhood adversity impacts on brain subcortical structures relevant to depression. *Journal of Psychiatric Research*, 86, 58–65. <https://doi.org/10.1016/j.jpsychires.2016.11.010>
- Goltermann, J., Meinert, S. L., Hülsmann, C., Dohm, K., Grotegerd, D., Redlich, R., Waltemate, L., Lemke, H., Thiel, K., Mehler, D. M. A., Enneking, V., Borgers, T., Repple, J., Gruber, M., Winter, N. R., Hahn, T., Brosch, K., Meller, T., Ringwald, K. G., ... Dannowski, U. (2023). Temporal stability and state-dependence of retrospective self-reports of childhood maltreatment in healthy and depressed adults. *Psychological Assessment*, 35(1), 12–22. <https://doi.org/10.1101/2021.08.31.21262884>
- Good, C. D., Johnsrude, I. S., Ashburner, J., Henson, R. N. A., Friston, K. J., & Frackowiak, R. S. J. (2001). A voxel-based morphometric study of ageing in 465 normal adult human brains. *NeuroImage*, 14(1), 21–36. <https://doi.org/10.1006/nimg.2001.0786>
- Hernandez, A., Gallardo-Pujol, D., Pereda, N., Arntz, A., Bernstein, D. P., Gaviria, A. M., Labad, A., Valero, J., & Gutiérrez-Zotes, J. A. (2013). Initial Validation of the Spanish Childhood Trauma Questionnaire- Short Form: Factor Structure, Reliability and Association With Parenting. *Journal of Interpersonal Violence*, 28(7), 1498–1518. <https://doi.org/10.1177/0886260512468240>
- Marek, S., Tervo-Clemmens, B., Calabro, F. J., Montez, D. F., Kay, B. P., Hatoum, A. S., Donohue, M. R., Foran, W., Miller, R. L., Hendrickson, T. J., Malone, S. M., Kandala, S., Feczko, E., Miranda-Dominguez, O., Graham, A. M., Earl, E. A., Perrone, A. J., Cordova, M., Doyle, O., ... Dosenbach,

- N. U. F. (2022). Reproducible brain-wide association studies require thousands of individuals. *Nature*, *603*, 654–660. <https://doi.org/10.1038/s41586-022-04492-9>
- McLaughlin, K. A., Weissman, D., & Bitrán, D. (2019). Childhood Adversity and Neural Development: A Systematic Review. *Annual Review of Delevopmental Psychology*, *1*, 277–312.
- Schaefer, J. D., Cheng, T. W., & Dunn, E. C. (2022). Review Sensitive periods in development and risk for psychiatric disorders and related endpoints: a systematic review of child maltreatment findings. *The Lancet Psychiatry*, *9*(12), 978–991. [https://doi.org/10.1016/S2215-0366\(22\)00362-5](https://doi.org/10.1016/S2215-0366(22)00362-5)
- Teicher, M. H., Anderson, C. M., Ohashi, K., Khan, A., McGreenery, C. E., Bolger, E. A., Rohan, M. L., & Vitaliano, G. D. (2018). Differential effects of childhood neglect and abuse during sensitive exposure periods on male and female hippocampus. *NeuroImage*, *169*, 443–452. <https://doi.org/10.1016/j.neuroimage.2017.12.055>
- Teicher, M. H., Samson, J. A., Anderson, C. M., & Ohashi, K. (2016). The effects of childhood maltreatment on brain structure, function and connectivity. *Nature Reviews Neuroscience*, *17*(10), 652–666. <https://doi.org/10.1038/nrn.2016.111>
- Wingenfeld, K., Spitzer, C., Mensebach, C., Grabe, H. J., Hill, A., Gast, U., Schlosser, N., Höpp, H., Beblo, T., & Driessen, M. (2010). The German Version of the Childhood Trauma Questionnaire (CTQ): Preliminary Psychometric Properties. *Psychotherapie Psychosomatik Medizinische Psychologie*, *60*, 442–450. <https://doi.org/10.1055/s-0030-1253494>
- Winter, N. R., Leenings, R., Ernsting, J., Sarink, K., Fisch, L., Emden, D., Blanke, J., Goltermann, J., Opel, N., Barkhau, C., Meinert, S., Dohm, K., Repple, J., Mauritz, M., Gruber, M., Leehr, E. J., Grotegerd, D., Redlich, R., Jansen, A., ... Hahn, T. (2022). Quantifying Deviations of Brain Structure and Function in Major Depressive Disorder Across Neuroimaging Modalities. *JAMA Psychiatry*, *79*(9), 879–888. <https://doi.org/10.1001/jamapsychiatry.2022.1780>

Major revision R2 of the manuscript “Gray matter correlates of childhood maltreatment: searching for replicability in a multi-cohort brain-wide association study” [NCOMMS-24-56581-B] to Nature Communications

We thank the reviewers once more for all constructive feedback which has helped us to once more improve our manuscript. We have added an additional complementary gray matter modality using parcellation-based measures of cortical thickness and surface, as well as subcortical volume to address concerns by the reviewers regarding the focus on single voxel overlap and regarding the utilization of VBM as a methodology. In addition, we have added mean global gray matter for each person as another dependent variable to explore the possibility of global CM effects on gray matter. Furthermore, we have added models investigating CM effects in medication-naïve MDD patients, as well as models including an interaction term between CTQ sum and age to explore the potential moderating role of age for CM effects. The additional analyses yielded results consistent with our previous null findings.

REVIEWER COMMENTS

Reviewer #1 (Remarks to the Author):

I appreciate the substantial effort the authors have made to address my previous concerns. However, I still have several critical issues that need to be resolved.

CTQ Scores and Variance Considerations

Including the CTQ sum score is a step in the right direction, but it is insufficient. To fully preserve the variance attributable to each type of maltreatment, the individual CTQ subscale scores should also be included, despite their intercorrelations. The harmonization process should not obscure the unique contributions of different types of maltreatment.

We would like to stress that the relevance of including ‘biological covariates’ in the Combat harmonization procedure is ultimately bound to the degree to which the variable in question is confounded with scanner groups. This can be particularly relevant when investigating small samples across different scanners where a model variable differs between scanner groups. As stated in our previous rebuttal our scanner groups only show minimal differences in CTQ sum (partial $\eta^2 = .003$). Accordingly, we demonstrated that the addition of CTQ sum as a biological covariate did not change our results. In fact, exemplarily comparing Table S5 from the original submission (completely neglecting the inclusion of CTQ sum as a biological covariate in the harmonization procedure) yielded the exact same results (even in decimal places) as the corresponding revised Table S8 (R1), for which CTQ sum was included as a biological covariate. The further refinement of the harmonization procedure by including individualized biological covariates depending on the specific predictor of a corresponding model will only impact results if this specific predictor shows a stronger confounding with scanner groups than CTQ sum (and that this confounding is not already accounted for by the other biological covariates included). This is in principle possible, however highly unlikely as the CTQ subscales and CTQ sum show high intercorrelations. To formally test this potential impact of including individual biological covariates, we investigated the association (i.e., confounding) between scanner groups and all CTQ subscales, while controlling for all biological covariates already previously included, thus testing the additional variance that scanner groups explain in CTQ subscales in addition to the previous procedure. This rationale tests if there is a residual confounding between scanner groups and CTQ subscales beyond

our current biological covariates. This ANCOVA yielded highly similar effect size estimates for the residual association between scanner groups and CTQ subscales as we obtained for CTQ sum (ranging from partial $\eta^2=.0001$ to partial $\eta^2=.001$; all non-significant). This means that there is no residual association (i.e., confounding) between CTQ subscales and scanner groups and consequentially no reason to believe that a further refinement of biological covariates would yield different results. Nonetheless, to fully address the reviewers concern, we examined the potential confounding effect by rerunning the harmonization procedure exemplary for Model 9 (non-sex-stratified) using the sexual abuse subscale of the CTQ instead of CTQsum as a biological covariate in Combat. The reason that we chose the sexual abuse subscale model for this test was that this subscale shows the lowest correlation with CTQsum, thus having the highest potential for added value in the harmonization process. As expected, this refined harmonization yielded almost identical results across all cohorts. To illustrate this, we compared voxel-wise t-values using the refined harmonization approach (CTQ sexual abuse subscale as biological covariate) and t-values resulting from the previous harmonization approach (CTQsum as biological covariate). These were virtually identical with the largest difference between voxel-wise t-values amounting to $1.02e-14$ (see Rebuttal Figure 1). We therefore conclude that residual confounding between scanner groups and CTQ subscales is unlikely to have introduced substantial bias into our results.

Rebuttal Figure 1

Scatter plot showing the association between brain-wide t-values for Model 9 (testing the CTQ sexual abuse subscale as a predictor), when including CTQ sum vs. CTQ sexual abuse as the biological covariate for childhood maltreatment in the Combat harmonization procedure

Note. The largest voxel-wise difference between t-values, comparing both harmonization approaches is given in each plot (Max Difference). CTQ_SA: sexual abuse subscale of the CTQ.

Limitations of Whole-Brain VBM Analysis

The use of whole-brain VBM to detect group differences is problematic, as only highly significant and widespread differences typically survive multiple comparison corrections at the whole-brain level.

Given the importance of this negative finding, I recommend complementing the whole-brain approach with region-of-interest (ROI) analyses. Specifically, previously implicated structures such as the hippocampus should be examined using selective masking to provide a more targeted test.

We agree that region-of-interest analyses are usually useful and common practice in order to reduce the number of statistical tests, thus improving the statistical power to find effects. This however only affects statistical thresholds derived when correcting for multiple testing. In our replicability analysis we applied uncorrected significance thresholds which are unaffected by the number of tests conducted. In other words, testing a hippocampus ROI would render the exact same voxels significant at $p\text{-uncorrected} < .01$ as would show in the same localization when testing at the whole-brain level. This makes the utilized thresholds of uncorrected $p < .001$ and $p < .01$ so liberal, further emphasizing the weight of the null-finding (particularly in often-cited regions, such as the hippocampus). Consequently, a ROI approach would only yield more significant findings in our pooled analysis if it leads to the corrected threshold of $p\text{-FWE} < .05$ being more liberal as compared to the uncorrected threshold of $p < .01$. To rule out this possibility we conducted an exemplary hippocampus ROI analysis to find out whether a $p\text{-FWE} < .05$ threshold in a ROI is more liberal or stricter as compared to an uncorrected $p < .01$ threshold. We used the aal atlas for the hippocampus ROI and reran model 13 (i.e., the model with the largest effects). We used the pooled sample for this (non-sex-stratified). The hippocampus ROI reduced the number of tests to $k=4259$. We obtained uncorrected and FWE-corrected p -values for this ROI analysis to compare these. The FWE-corrected p -value closest to $p = .05$ corresponded to an uncorrected p -value of $p\text{-unc} = .0005$. In contrast, the uncorrected p -value closest to $p = .01$ corresponded to an FWE-corrected p -value of $p\text{-FWE} = .3433$. The relationship between corrected and uncorrected p -values in this ROI analysis are illustrated in Rebuttal Figure 2. Our results further support our rationale to refrain from conducting further ROI analyses. The relationship correspondence between corrected and uncorrected p -values in this ROI analysis are shown in Rebuttal Figure 2. Despite our consistent whole-brain approach, we have added complementary analyses using segmentation-based regional measures of cortical thickness and surface, as well as subcortical volume via Freesurfer, which implicates a substantial reduction of statistical tests, thereby increasing statistical power for regions such as the hippocampus.

Rebuttal Figure 2

Scatter plot showing the association between uncorrected p-values and FWE-corrected p-values in an exemplary region-of-interest analysis in the hippocampus in Model 13 using the pooled sample, non-sex-stratified.

Concerns Regarding Overlapping Voxels

The emphasis on exactly overlapping voxels in VBM analysis is not particularly informative. The VBM pipeline involves morphing individual brains to a standard template, meaning that a given voxel in the template corresponds to slightly different anatomical locations across individuals. A more meaningful approach—consistent with meta-analytic methods—would be to assess effect sizes for maximal differences within specific regions, allowing for a more robust evaluation of effect sizes and study heterogeneity.

We agree that there likely is interindividual variation in the exact localization of CM effects. The hypothesized impact that CM has on the brain presumably is not precisely in the same voxel for each person, let alone in one voxel alone. However, our underlying rationale does not make these assumptions. We do not assume that one specific voxel has to be significant for every single person but rather that a cluster of voxels is significant on a group-level and this cluster is overlapping with the cluster on a group-level in another sample/cohort. The extent of this overlap is defined as the number of voxels showing significance in both samples. The clusters identified in our cohort-wise analysis show little to no overlap which is interpreted as meaningful non-replicability. While interindividual gray matter reductions could vary in their spatial localization the common theory is that there is between-subject convergence in this localization on a group-level (e.g., effects localized in specific regions, such as the hippocampus) – a notion that could potentially be challenged. Particularly, in large samples like ours such a spatial convergence should become evident even when interindividual variance in the exact localization is present (e.g., when an effect is found a centimeter apart in the hippocampus in another person). Thus, we would argue that the rationale of focusing on voxel-wise spatial overlap in significance is still a valid approach. However, we agree that it is a possibility that gray matter correlates of CM could be present, while not showing spatial convergence on a group-level. This notion is in part reflected by our analysis on global gray matter density described below as a reply to the reviewer's critique on correcting for TIV. The possibility of individual gray matter deviations from the norm as a consequence of CM, which can lack spatial convergence on a group-level is shortly discussed within the discussion section (i.e., normative modelling; lines 417 ff.).

While we think that our voxel-wise analysis is a valid approach, we decided to additionally use aggregated measures of cortical thickness/surface and subcortical volume to address the reviewer's concern that the focus on exact voxel overlap may be misleading. This was done using a complementary operationalization of gray matter structure using the Freesurfer software. Desikan-Killiany-parcellated cortical measures for regional thickness and surface, as well as regional subcortical volumes were obtained and used as a dependent variable in the same models as in the voxel-wise approach. Parcellation-based results are now described in lines 272 ff. in the main text and are disclosed in more detail within supplementary tables S19 and S20. The applied methods are described shortly in the methods section of the main text in lines 619 ff. and in more detail within the supplements in lines 193 ff.

An additional meta-analytic approach would mean to aggregate the summary statistics of each cohort to obtain an estimate for the joint evidence for an effect. This is commonly done to synthesize results from different studies when only summary statistics are available. As we have full access to all individual datapoints a mega-analysis is preferable to a meta-analysis due to increased precision in the obtained statistical estimates for mega- as compared to meta-analysis (Boedhoe et al., 2019). Our analysis using the pooled cohorts closely resembles the approach of such a mega-analysis. We however have openly provided non-thresholded statistical brain maps for all analyses (across each cohort separately) in our online OSF repository, thus enabling future meta-analytic studies to fully take into account the evidence from our datasets.

Potential Age Confound

Age may be a confounding factor in the reported findings. The effects of maltreatment-related brain alterations may be more discernible in younger adults than in older adults due to neurodevelopmental trajectories. I suggest conducting age-stratified analyses to determine whether these effects are more pronounced within specific age bands.

*It is true that some previous studies have noted the potential moderating role of age for maltreatment effects on the brain. We have carefully considered the reviewer's suggestion and think that this is of interest, albeit challenging. We agree that potential age interactions are of interest and could be potentially relevant. On the other side, it is difficult to validly investigate this in a comprehensive manner, which could easily constitute an independent research project by itself. The reviewer suggests to investigate specific age bands, thus stratifying the sample for different age groups. The problem we see here, is that there is little knowledge about age borders where effects of CM would be different, thus providing a poor basis for grouping rules (how many groups, at what cutoffs etc.). Furthermore, we have already done extensive analyses prior to this revision already spanning 15 different statistical models, in 3 cohorts plus in the pooled cohort, as well as sex-stratified analyses, summing up to 180 voxel-based whole-brain level analyses. We furthermore have added the same 180 analyses investigating a total of 154 parcellation-based gray matter measures to address some of the reviewers' concerns. At this point it would not be not feasible to repeat all analyses again within multiple age bands. For example, investigating three different age bands would effectively quadruple the number of analyses, while reducing the sample sizes within each analysis, further compromising statistical power. We believe that such an enormous number of analyses would not be beneficial to convey the main findings of this manuscript. However, despite these difficulties we believe that the inclusion of age-dependent CM effects constitutes a valuable addition to our project and have decided to investigate CTQ sum * age interaction terms within HC and MDD subsamples separately (similar to model 3 and model 4 but with an interaction term). After careful consideration we have decided to refrain from investigating interaction terms also in the other models for different reasons: Model 1 and model 2 are both criticizable for their confounding with MDD group (either underestimating CM effects by controlling for covarying MDD in model 1 or overestimating the CM effect due to neglecting confounding MDD in model 2). Furthermore, the other models already investigate distinct specialized cases under which CM effects may become evident (similar to investigating a moderating effect of different circumstances), such as specific subtypes of CM (models 5-11) or extreme forms of CM (models 12-15). We believe that the additional investigation of age interactions within HC and MDD subgroups provide the best information regarding the potential role of age, while keeping the number of analyses at a reasonable amount. We have added a short introduction into previous evidence for age interactions and have added two additional models (Model 17, HC; Model 18, MDD) which we now ran again for all cohorts and sex-subgroups, as well as for both complementary gray matter modalities (voxel-wise and parcellation-based). The results yield similar non-replicability as all models probing main effects of CM. The introduction was added in line 116, the results were added to all respective results tables and shown in figures S22-23; S40-41; S58-59, and the statistical models are described in lines 607 ff. of the main manuscript.*

Medication Status as a Confound

The manuscript does not address the medication status of participants, which could significantly influence the results. Many studies reporting large effects of childhood maltreatment have

specifically examined unmedicated samples. It is important to control for medication use in the analysis and, if possible, conduct a separate analysis limited to unmedicated participants.

We thank the reviewer for this important remark. We agree that medication can have an effect on brain structure when investigating psychiatric patients and should thus be addressed, although meta-analytic evidence suggests that no additional CM effects on gray matter are present in non-medicated patients (Lim et al., 2014). We have now added a description of the distribution of participants taking antidepressant medication across included cohorts in Table S1. These descriptives reveal that only small subsamples of unmedicated patients are available in our cohorts (particularly within the MNC and BiDirect cohorts), severely limiting the interpretability of replicability analyses in these subsamples. Nevertheless, we conducted additional analysis in unmedicated MDD patients only, in our new Model 16 (similar to model 4 but only including MDD patients not taking antidepressant medication). This model yielded results very similar to model 4 investigating all MDD patients. We have added a short section carefully discussing these findings with emphasis on the limitations due to the small sample sizes in lines 389 ff. In this passage we also elaborate on the possibility of medication effects on the brain potentially masking CM effects, thus causing our observed null finding. For this, we reference a meta-analysis on CM effects on gray matter, which conducted additional sensitivity analysis in non-medicated patients, not reporting on additional CM-associated findings in this analysis (Lim et al., 2014).

Limitations of VBM as an Analytic Tool

VBM represents a relatively small proportion of published neuroimaging studies on childhood maltreatment, comprising roughly 15-20% of the literature. Prior analyses, such as those in Teicher et al. (2016), have demonstrated that VBM studies tend to yield less consistent findings compared to other neuroimaging methods. Our research group has discontinued the use of VBM for studying maltreatment-related brain changes due to these limitations. The authors should explicitly acknowledge these limitations in the discussion.

We thank the reviewer for this additional remark on the utilized gray matter operationalization and methodology. While this may not be representative for the complete literature, as far as we can see previous meta-analyses on gray matter effects of CM do not indicate a substantial inferiority of VBM studies, neither in the number of studies, nor in their sensitivity to detect significant effects (Paquola et al., 2016; Yang et al., 2023). Furthermore, thoroughly checking two publications by Teicher et al. from 2016 (Teicher et al., 2016; Teicher & Samson, 2016) we were not able to find any explicit indication for less consistent findings in VBM findings. We apologize if we may have misunderstood the reference or if the reviewer is referring to a different piece of evidence. We are not aware of any evidence that points to a methodological inferiority of VBM studies. We are aware that there is an ongoing debate about what is captured by VBM studies vs. in parcellation-based approaches and to what extent findings are comparable (Goto et al., 2022). In our opinion the advantage of VBM is the high spatial resolution that could lead to a higher sensitivity particularly in cases, where the aggregation of regional gray matter for one area may blur results (e.g., in one portion of the hippocampus different effects than in another portion). The main advantage of parcellation-based measures of gray matter is the easier data handling and anonymization of data, thus enabling research groups to share and pool data (as done in the ENIGMA consortium). Furthermore, as noted by the reviewer the segmentation-based measures are less subject to data transformations during the normalization preprocessing steps. Overall, both methodologies can be seen as complementary, likely to be measuring slightly different neurobiological entities. Thus, we have decided to conduct additional parcellation-based analyses based on Freesurfer measures of cortical thickness and surface, as well as subcortical

volume, as described above. The addition of this gray matter modality yielded results consistent with our voxel-based analyses, thus further strengthening our null finding. Within the pooled analysis the largest effect size was found for model 2 for the thickness of the right middle temporal gyrus with partial $R^2=.005$. Thus, we found no evidence for larger effects (or more consistent effects) in parcellation-based analysis as compared to VBM analysis. Parcellation-based results are now described in lines 272 ff. in the main text and are disclosed in more detail within supplementary tables S19 and S20. The applied methods are described shortly in the methods section of the main text in lines 619 ff. and in more detail within the supplements in lines 193 ff.

Implications of Total Intracranial Volume (TIV) Correction

The decision to control for TIV warrants further consideration. While TIV correction is generally necessary to account for individual differences in brain size, it assumes that TIV is independent of pathology. In some conditions, such as neurodevelopmental disorders (e.g., autism, ADHD), smaller TIV is an intrinsic feature rather than a confound. In cases where widespread reductions in gray matter volume have been reported (e.g., De Bellis et al.), TIV correction may lead to overcorrection, potentially obscuring true volumetric differences. This issue is particularly relevant when assessing the effects of severe early neglect.

We apologize but we were not able to identify the specific publication by De Bellis you were referring to. We agree that global measures of gray matter could potentially be impacted by CM without regional/localized effects showing in our brain-wide analyses. Our newly added parcellation-based analyses also include regional measures of cortical thickness for which we did not control for TIV as these cortical thickness measures are generally less confounded with head size. Apart from that we think that the idea of a global CM effect on gray matter which could be masked by including a variable such as TIV is an interesting possibility. As TIV is the intracranial volume (i.e., head size), we believe this is not the most suitable target variable to investigate this. Instead, we have decided to calculate the average gray matter density across all voxels per person (using the normalized VBM images). These individual mean gray matter densities constitute the global gray matter volume of each person (gray matter intensities per voxel of the normalized images represent gray matter volume in the non-normalized images). We investigated this global gray matter variable as an additional brain variable together with our parcellation-based variables. In these models we did not additionally control for TIV. While we found a negative association between this global gray matter density variable and CM in several models in the pooled analysis, we could not confirm any replicability of this association across single cohorts. We have added a short description of this result in lines 276 ff. and 283 ff., as well as a description of the methodological approach within the main text in lines 622 ff. and disclose results in more detail within the supplements in Table S19.

Overall, I recommend that the authors carefully reconsider these issues to strengthen the rigor and interpretability of their findings.

Reviewer #2 (Remarks to the Author):

The authors have thoroughly and adequately responded to my comments. I have no further recommendations.

We want to take the opportunity to thank the reviewer for the helpful remarks and the benevolent assessment.

Reviewer #3 (Remarks to the Author):

1. The authors have done a much better job describing their cohorts, in particular with respect to levels of CM and depression, and their correlation. However, their writeup still overstates the ability of the analysis to disentangle psychopathology and CM in their sample. Average CTQ scores in the HC group were close to the floor of the instrument, and the rank-biserial r between MDD status and CTQ sum was above .5 in every cohort -- an exceptionally strong confound. If this were a probability sample, this correlation could be considered a relatively natural phenomenon, but the fact that each sample was recruited to over-sample depressed individuals likely inflates this correlation (because the CM we are observing is mostly contingent on someone being in the MDD group), a problem which is further compounded by concerns that the retrospective CM measure could be contaminated by reporting biases associated with symptoms. In their response, the authors argue that this confounding would be more likely to create a false positive than false negative, and I am not sure why. If the model is controlling for a confound correlated with the predictor at .5, it is very likely that the effect of the predictor will be underestimated and/or unstable. They further argue that diagnosis likely drives CM effects because the effects are not visible in their subgroup (within HC, within MDD) analysis. However, they did report effects in their pooled sample in the HC group, despite the fact that their power was greatly limited by the limited amount of CM variability in this group. The absence of an association of CM with gray matter structure in the MDD group is interesting.

In light of the strength of the confounding and the limitations of the study design, my opinion is that the models controlling for MDD diagnosis are of limited interest and should be omitted.

We agree that the confounding between CM and depression constitutes a challenge to plan suitable analyses to investigate neural correlates of CM and to interpret findings. In our opinion, all models including HC and MDD participants have limited interpretability either due to putative overestimation of a 'CM effect' when not controlling for MDD diagnosis or due to underestimation of the 'CM effect' when controlling for MDD diagnosis. In our last rebuttal we were referring to Model 4 (MDD samples only) when stating that residual confounding between CM and depression could lead to false positive findings, assuming alleged CM effects could be driven by confounding depression severity (even within MDD samples due to variability in depression severity and previous course of disease, both associated with CM). We apologize for this misunderstanding. The main argument that we see for the 'CM effects' in Model 2 and Model 13 likely being attributable to neural correlates of depression is, that dedicated investigations of MDD case-control differences in another paper using the same cohorts lead to highly similar clusters (Dannlowski et al., 2024) and that both HC and MDD subgroups do not show any evidence for replicability of CM effects (despite some HC models yielding significant results within the pooled analysis).

If we have understood correctly, further main points by the reviewer are, that 1) high confounding between depression and CM are worsened by our sampling strategy, 2) low CM variability in HC samples may compromise statistical power in replicability analysis, 3) there could be a depressive reporting bias of CM and that 4) models controlling for MDD diagnosis should be omitted from the manuscript.

Regarding our sampling strategy, we agree that including MDD samples always comes with the additional challenge to disentangle naturally confounded depression and CM effects, and this problem is worsened when not investigating a representative sample. We have now emphasized this limitation in our discussion section in lines 369 ff.

The notion that low statistical power in the HC subsamples is not only caused by small samples but also aggravated by a suboptimal statistical distribution of CTQ sum scores in this

group is interesting. We have added an elaboration on this within the discussion section in lines 389 ff., basically adopting the reviewer's position on this up to the point that the statistical power is likely limited by this. However, we also highlight that such low predictor variability is not substantially different from other previous studies that report on significant CM effects in non-clinical populations.

With regard to a potential depressive reporting bias we have described evidence from the MACS and MNC samples suggesting against such a bias in these cohorts (Goltermann et al., 2023). However, despite these findings we agree that it is difficult to fully exclude such biases. Thus, we have once more emphasized the potential of a depressive memory/reporting bias in lines 333 ff.

We fully understand the reviewer's notion that the interpretability of the models controlling for MDD diagnosis is limited due to the strong association between CM and MDD. However, we would like to refrain from omitting the conducted analyses as they mirror a frequently adopted approach in previous studies investigating the matter (e.g. Chaney et al., 2014; S. Lu et al., 2019; X. W. Lu et al., 2018). However, we have now revised the discussion section to put more emphasis on the limited interpretability of these models as described above (lines 369 ff.). We would even argue that a null finding when controlling for MDD can be consistent with theories of how neurobiological alterations convey the clinical consequences of CM experiences. This assumption can be illustrated as a mediation model with neurobiological alterations being the mediator between CM and the psychiatric consequences. In such a mediation model we would even expect the statistical association between CM and neural correlates to be diminished or to disappear when controlling for depression due to expected shared variance. The only clean way to validly disentangle these effects would be a long-term longitudinal study, assessing CM prospectively during childhood/adolescence and assessing long-term trajectories of clinical and neurobiological developments. We have added this notion also within the discussion section in lines 352 ff.

2. Marek et al. (2022) concluded a sample of 9500 would be required to detect a relatively strong brain-wide-association-study effect, 80% of the time, with a corrected p of $p < 10^{-7}$ (which is still more lenient than $.001^3$), and that a sample of 2200 would be required to adequately detect a large effect with an uncorrected $p < .05$. They defined 'replicability' as 'passing the same statistical threshold in sample and out of sample'. In their replication test, the authors' are using an alpha value of $.001$ (which equates to an effective false negative rate of $.001^3$ or $p \leq 10^{-10}$) with cohort n 's between 557 and 1752. Rather than test each cohort against the others (in sample vs out of sample), they chose to test the overlap between all three cohorts. Could you please better justify the alpha levels defined in the replicability analysis, as well as the choice to compare overlap between three samples rather than out of sample vs in sample for each cohort? As it stands, the lack of replicability you are reporting is easily explained as a power problem in the context of a rather strict alpha threshold. This is still potentially interesting given the even smaller n 's common in the literature you are critiquing (although those are usually paired with more liberal alpha levels), but these choices need justification and discussion.

This is a very important point, as no consensus exists for most valid significance thresholds (or generally replication success criteria) for replications of neuroimaging findings, particularly in voxel-based analysis. The matter is complicated by the mass-univariate testing and the lack of valid power analyses in this domain. It is true that the significance thresholds utilized here are somewhat arbitrary. The most common definition of replication success in replication research is finding the same effect again at the same significance threshold as used in the original study (for analyses with one dependent variable mostly $p < .05$). Even in the simplest

case with only one statistical test a replication analysis applying the same threshold as the original analysis already leads to an accumulated stricter statistical threshold overall. In other words, the accumulated alpha error is lower for finding something in an original study and again in a replication study than only finding something once. The same is true for the statistical power which behaves complementary (the probability of finding a true effect is lower when applying the same statistical threshold again in a new sample). This phenomenon is inherent to replication research. We believe that generally a natural starting point for any replication study is to use the original sample's threshold again for the replication sample. Thus, we originally started off with applying an FWE-corrected threshold of $p < .05$ for all replication analyses due to this being a standard approach when conducting voxel-wise analysis in an original study. Notably, this approach, while arguably being very conservative when applying FWE correction for a whole-brain analysis, can already lead to the detection of successfully replicable neural correlates within the realm of psychiatric conditions: we used a similar approach as in the manuscript at hand to investigate the replicability of MDD case-control differences on gray matter and found that the most robust effects replicated across all pairwise cohort combinations between MACS, MNC and BiDirect even at an FWE corrected $p < .05$, even with small underlying effect sizes reaching a maximum of partial $R^2 = .01$. While replication studies are very rare in neuroscience, one other study investigating functional connectivity correlates of emotion regulation used a similar approach, applying an FWE-corrected threshold of $p < .05$ for a replication (Dörfel et al., 2020). However, we acknowledge that a null finding at such a strict threshold applied at multiple cohorts is indeed conservative and the interpretability somewhat difficult. Additionally, studies may originally apply such strict thresholds within a ROI analysis in order to minimize the number of statistical tests to boost statistical power. That is why we decided to move towards more liberal thresholds when originally not finding any voxel overlap using $p_{FWE} < .05$. This resulted in us checking for overlap in significance at uncorrected $p < .001$ (which is a common threshold for liberal exploratory whole-brain results) and finally in $p < .01$ which is one magnitude more liberal than what would be normally used to obtain whole-brain results. As we have outlined above as a reply to one of the critiques of reviewer 1 an uncorrected p-value of $p = .01$ corresponds to an FWE-corrected p-value of $p_{FWE} = .3433$ when conducting a hippocampus ROI to limit the search space (see also Rebuttal Figure 2). Thus, even when increasing the statistical power in a ROI approach such a threshold would hardly be used for statistical inference. This comparison was undertaken to facilitate interpretability of the conservatism/liberalism of the applied thresholds. How liberal thresholds should be to validly investigate replicability is arguably up for debate. Our permuted null distributions give some contextual information about the bottom-line conservatism/liberalism of the applied thresholds when used in combination with multiple cohorts. For example, the histogram below (Rebuttal Figure 3) shows the distribution of pairwise overlap happening by chance, permuting model 2, using either a cohort-wise threshold of uncorrected $p < .01$ or $p < .001$ (using the MACS-MNC example for pairwise overlap). The figure illustrates that using $p < .001$ rarely leads to pairwise overlap by chance, with small numbers of voxels overlapping in the non-permuted run already having a good chance becoming significant. In contrast, overlap can be produced by chance using $p < .01$ more often, leading to voxel overlap of hundreds or thousands of voxels by chance (in this example meaning that even the observed overlap of $k = 73$ can be non-significant).

In summary, even though the utilized thresholds are somewhat arbitrary as no convention exists for replications, we believe that we have learned something about their usefulness to define replicability: In large samples highly replicable effects (even with small effect sizes at partial $R^2 \sim .01$) can be identified even using a conservative threshold, such as $p_{FWE} < .05$ as can be seen in our MDD case-control analysis (Dannlowksi et al., 2024). Furthermore, more liberal thresholds can be used to provoke significance overlap by chance even for underlying null effects and compare the observed overlap with a permuted null distribution. When combining the replicability analysis with such a permutation-based significance test, even a liberal

threshold (i.e., $p < .01$) can be used to estimate the above-chance replicability of voxel-wise effects. We hope that our efforts to operationalize replicability will help inform and further investigate future definitions of replicability of voxel-based findings.

Despite the inherent increase in conservatism when conducting replication research (due to the combination of thresholds across samples, as outlined above), we agree that even the uncorrected $p < .001$ threshold is very conservative when applied to three cohorts as calculated by the reviewer. However, in contrast pairwise overlap at $p < .01$ was also investigated, leading to a far more liberal accumulated threshold of $.01 * .01 (= .0001)$ which would not be considered strict when conducting voxel-wise analyses (i.e., over 400000 mass-univariate tests). This info regarding the resulting effective false-positive when applying $p < .01$ to two cohorts was added in lines 559-560. The reviewer references the study by Marek et al. to argue that our approach could be underpowered. We agree that for small effect sizes (e.g., partial $R^2 < .01$) our sample sizes could be underpowered, as we have discussed as a limitation within our discussion section. However, we would like to point out that the Marek power calculations are not directly applicable to our analyses. Conventional power calculations are ultimately depending on the alpha-error (i.e., p-value threshold), the sample size and the effect size. However, conducting a high number of tests substantially complicates this. In the study by Marek et al. segmentation-based brain variables were used (e.g., $n=333$ regional brain variables for cortical thickness). For voxel-wise analyses the nature of the mass-univariate testing (~ 400000 voxels/tests) leads to higher possibility to find an effect (both, by chance or real effect). Thus, the false-negative rate can only be interpreted for 1 given test but it is unclear to what extent this conclusion can be generalized for finding an effect in a series of 400000 tests. Conducting more tests, increases the probability of finding a true effect (i.e., power) but also increases the probability of false-positives. To what extent power and false-positives are increased depends on the degree of spatial dependencies of the tests. As the covariance structure of the voxel-wise data is difficult to assess (depending on anatomical peculiarities, but also smoothing and the nature of the effect being localized or global), power analysis in voxel-wise data is not trivial and despite a number of tools attempting to approach this problem for example by simulation techniques, none of these have gained broad acceptance in the field. In fact, many attempts for this have been discontinued over the past years. The power calculations conducted by Marek et al. are only validly generalizable when using a similar number of tests with a similar spatial covariance structure as in their study, which is arguably not the case in our voxel-wise analysis. Furthermore, the analysis by Marek et al. included non-clinical cohorts only and it has been argued that effect sizes could possibly be larger in clinical samples. In fact, the effect sizes of MDD diagnosis and of CM in our two studies seem to be partial $R^2 \sim .01$. This is approximately three times as large of an effect as compared to the largest effect size observed in the investigation by Marek et al. (there, top 1% effect sizes $r > .06$, corresponding to $R^2 > .0036$), which arguably has a substantial impact on power calculations. Lastly, it is important to consider that we applied one-sided negative contrasts, thus effectively halving our p-values towards becoming considerably more liberal when compared with two-sided p-values (Marek et al. used two-sided p-values). With regard to the number of conducted tests, our newly added analysis of segmentation-based regional Freesurfer measures of gray matter structure is probably more comparable to the power analysis of conducted by Marek et al. (although, here we use fewer brain measures in comparison). In the segmentation-based analysis we applied an FDR-corrected threshold of $p < .05$ with a similar null finding as in previous analyses. As our discussion section is already quite extensive, we would like to refrain from including a more detailed elaboration on the applied significance thresholds and their implications for statistical power than currently included (currently stating that our study could potentially be underpowered for very small effect sizes, particularly in certain subsamples). A more detailed discussion of this would require a rather lengthy argumentation including the arguments outlined above, which we

feel is outside the scope of the current manuscript, despite the general relevance of this matter.

As to our approach to investigate each cohort separately and investigate their overlap, we think that this fits best to our research question which was focused on cohort-wise replicability. In the study by Marek et al. the referenced replication analysis was conducted based on a resampling procedure, thus drawing different size-matched original and replication samples (i.e., the 'in sample' and 'out of sample' samples), while manipulating sample size. This approach resembles our pairwise comparisons, with the difference that we used sample definitions predefined by the original study cohorts. If we understand the reviewer's remark correctly, he/she is suggesting that we could alternatively investigate replicability by pooling two cohorts and look for replicability of the obtained effects in the corresponding third cohort to increase statistical power. While this would increase statistical power of the discovery sample, it would not have any impact on the power of the respective replication sample. Furthermore, we found it more useful to compare all three cohorts separately, as this more closely mirrors the nature of the data as they represent three different studies. These studies, while collecting similar data in harmonized ways, differ from each other in some aspects, such as higher age in the BiDirect sample, or inpatient treatment recruitment in the MNC sample). Thus, overlap between any pairs of cohorts while not found for other cohort pairs could have been informative. For example, if overlap would have been found between MACS and MNC but not with either of these cohorts and BiDirect this could be due to sample characteristics specific to the BiDirect cohort (e.g., higher age). Pooling cohorts to form discovery samples would hamper this interpretability.

Rebuttal Figure 3

Permutation-based null distribution of pairwise and triple-wise overlap in significance at uncorrected $p < .01$ and $p < .001$

Minor

- The authors have introduced a new permutation test to test the significance of their replication findings. Could you please provide more details about this test explain why it was added post-hoc?

We are happy to provide additional details and rationale for our permutation test. As outlined above there is no consensus on how to define replication success when conducting voxel-wise replication studies. There is a variety of replication success definitions available (Open Science Collaboration, 2015), however the number of statistical tests and consequential need to correct for multiple testing, as well as the three-dimensional nature of MRI findings complicate the matter. As the reviewer has pointed out and as we have discussed above the selection of significance thresholds is somewhat arbitrary and it is unclear which thresholds are most useful when investigating voxel-wise replicability. The implemented permutation-based significance test is thought to formalize this by quantifying the extent to which overlap can be expected by chance (no matter how liberal the original significance threshold is). The idea for this approach was generated during the revision process of the manuscript and thus added post-hoc. We believe that this permutation test can be a straightforward way to estimate the utility of different significance thresholds for replications and to complement the mere descriptive number of voxels overlapping by a formalized significance estimate for the observed replicability (and thus increase interpretability). To obtain the null distributions we

ran the code of all models across all cohorts with permuted predictor columns (e.g., in Model 1 CTQ sum was permuted, in Model 15 the extreme CM group labels were permuted). This was repeated $n=1000$ times and then these 1000 permutation results were saved as thresholded brain maps at uncorrected $p<.01$ and $p<.001$. For each pairwise and triple-wise cohort combination the overlap in significance of these by-chance effects was calculated iterating through these $n=1000$ thresholded permutation results, thus creating a null distribution of replicability (i.e., voxel overlap for every given cohort combination for each model happening by chance). The observed overlap of the non-permuted run for each analysis was then compared to the obtained null distribution and the p -value calculated by taking the proportion of permuted runs that lead to an overlap larger than the observed overlap. This procedure was conducted across all cohort combinations and for all models thus creating a situation with multiple statistical tests. Consequently, we corrected the obtained overlap p -values using the Benjamini-Hochberg FDR correction for sets of models and cohort combinations (18 models multiplied by three different cohort combinations plus the triple-wise cohort combination – $n=72$ tests included in each FDR correction).

- In the limitations 'Although depressive states are thought to bias childhood reports,' I believe should be 'depressive states are thought to bias retrospective reports of childhood maltreatment'.

We thank the reviewer for bringing our attention to this mistake. We have adopted the suggestion.

References for the rebuttal:

- Boedhoe, P. S. W., Heymans, M. W., Schmaal, L., Abe, Y., Alonso, P., Ameis, S. H., Anticevic, A., Arnold, P. D., Batistuzzo, M. C., Benedetti, F., Beucke, J. C., Bollettini, I., Bose, A., Brem, S., Calvo, A., Calvo, R., Cheng, Y., Cho, K. I. K., Ciullo, V., ... Twisk, J. W. R. (2019). An Empirical Comparison of Meta- and Mega-Analysis With Data From the ENIGMA Obsessive-Compulsive Disorder Working Group. *Frontiers in Neuroinformatics*, *12*, 102. <https://doi.org/10.3389/fninf.2018.00102>
- Chaney, A., Carballedo, A., Amico, F., Fagan, A., Skokauskas, N., Meaney, J., & Frodl, T. (2014). Effect of childhood maltreatment on brain structure in adult patients with major depressive disorder and healthy participants. *Journal of Psychiatry and Neuroscience*, *39*(1), 50–59. <https://doi.org/10.1503/jpn.120208>
- Dannlowski, U., Winter, N. R., Meinert, S., Grotegerd, D., Kraus, A., Flinkenflügel, K., Leehr, E., Böhnlein, J., Borgers, T., Fisch, L., Bauer, M., Pfennig, A., Richter, M., Opel, N., Repple, J., Gruber, M., Minnerup, H., Hermesdorf, M., Nitsch, R., ... Goltermann, J. (2024). Replicability and generalizability of gray matter reductions in major depression: a voxel-based investigation of 4021 individuals. *SSRN (Preprint)*. <https://doi.org/http://dx.doi.org/10.2139/ssrn.4854882>
- Dörfel, D., Gärtner, A., & Scheffel, C. (2020). Resting State Cortico-Limbic Functional Connectivity and Dispositional Use of Emotion Regulation Strategies: A Replication and Extension Study. *Frontiers in Behavioral Neuroscience*, *14*, 1–14. <https://doi.org/10.3389/fnbeh.2020.00128>
- Goltermann, J., Meinert, S. L., Hülsmann, C., Dohm, K., Grotegerd, D., Redlich, R., Waltemate, L., Lemke, H., Thiel, K., Mehler, D. M. A., Enneking, V., Borgers, T., Repple, J., Gruber, M., Winter, N. R., Hahn, T., Brosch, K., Meller, T., Ringwald, K. G., ... Dannlowski, U. (2023). Temporal stability and state-dependence of retrospective self-reports of childhood maltreatment in healthy and depressed adults. *Psychological Assessment*, *35*(1), 12–22. <https://doi.org/10.1101/2021.08.31.21262884>
- Goto, M., Abe, O., Hagiwara, A., Fujita, S., Kamagata, K., Hori, M., Aoki, S., Osada, T., Konishi, S., Masutani, Y., Sakamoto, H., Sakano, Y., Kyogoku, S., & Daida, H. (2022). Advantages of Using

- Both Voxel-and Surface-based Morphometry in Cortical Morphology Analysis: A Review of Various Applications. *Magnetic Resonance in Medical Sciences*, 21(1), 41–57.
<https://doi.org/10.2463/mrms.rev.2021-0096>
- Lim, L., Radua, J., & Rubia, K. (2014). Gray Matter Abnormalities in Childhood Maltreatment: A Voxel-Wise Meta-Analysis. *American Journal of Psychiatry*, 171, 854–863.
- Lu, S., Xu, R., Cao, J., Yin, Y., Gao, W., Wang, D., Wei, Z., Hu, S., Huang, M., Li, L., & Xu, Y. (2019). The left dorsolateral prefrontal cortex volume is reduced in adults reporting childhood trauma independent of depression diagnosis. *Journal of Psychiatric Research*, 112(139), 12–17.
<https://doi.org/10.1016/j.jpsychires.2019.02.014>
- Lu, X. W., Guo, H., Sun, J. R., Dong, Q. L., Zhao, F. T., Liao, X. H., Zhang, L., Zhang, Y., Li, W. H., Li, Z. X., Liu, T. B., He, Y., Xia, M. R., & Li, L. J. (2018). A shared effect of paroxetine treatment on gray matter volume in depressive patients with and without childhood maltreatment: A voxel-based morphometry study. *CNS Neuroscience and Therapeutics*, 24(11), 1073–1083.
<https://doi.org/10.1111/cns.13055>
- Open Science Collaboration. (2015). Estimating the reproducibility of psychological science. *Science*, 349(6251), aac4716. <https://doi.org/10.1126/science.aac4716>
- Paquola, C., Bennett, M. R., & Lagopoulos, J. (2016). Understanding heterogeneity in grey matter research of adults with childhood maltreatment: A meta-analysis and review. *Neuroscience and Biobehavioral Reviews*, 69, 299–312. <https://doi.org/10.1016/j.neubiorev.2016.08.011>
- Teicher, M. H., & Samson, J. A. (2016). Annual Research Review: Enduring neurobiological effects of childhood abuse and neglect. *Journal of Child Psychology and Psychiatry*, 57(3), 241–266.
<https://doi.org/10.1111/jcpp.12507>
- Teicher, M. H., Samson, J. A., Anderson, C. M., & Ohashi, K. (2016). The effects of childhood maltreatment on brain structure, function and connectivity. *Nature Reviews Neuroscience*, 17(10), 652–666. <https://doi.org/10.1038/nrn.2016.111>
- Yang, W., Jin, S., Duan, W., Yu, H., Ping, L., Shen, Z., Cheng, Y., Xu, X., & Zhou, C. (2023). The effects of childhood maltreatment on cortical thickness and gray matter volume: A coordinate-based meta-analysis. *Psychological Medicine*, 53, 1681–1699.
<https://doi.org/10.1017/S0033291723000661>